# Towards Editing Time Series

**Baoyu Jing**[*2], **Shuqi Gu**[*1], **Tianyu Chen**[1], **Zhiyu Yang**[1],
**Dongsheng Li**[3], **Jingrui He**[2], **Kan Ren**[†1]
[1]ShanghaiTech University, [2]University of Illinois at Urbana-Champaign, [3]Microsoft Research

## Abstract

Synthesizing time series data is pivotal in modern society, aiding effective decision-making and ensuring privacy preservation in various scenarios. Time series are associated with various attributes, including trends, seasonality, and external information such as location. Recent research has predominantly focused on random unconditional synthesis or conditional synthesis. Nonetheless, these paradigms generate time series from scratch and are incapable of manipulating existing time series samples. This paper introduces a novel task, called Time Series Editing (TSE), to synthesize time series by manipulating existing time series. The objective is to modify the given time series according to the specified attributes while preserving other properties unchanged. This task is not trivial due to the inadequacy of data coverage and the intricate relationships between time series and their attributes. To address these issues, we introduce a novel diffusion model, called TEdit. The proposed TEdit is trained using a novel bootstrap learning algorithm that effectively enhances the coverage of the original data. It is also equipped with an innovative multi-resolution modeling and generation paradigm to capture the complex relationships between time series and their attributes. Experimental results demonstrate the efficacy of TEdit for editing specified attributes upon the existing time series data. The project page is at https://seqml.github.io/tse.

## 1 Introduction

Time series data analysis plays a crucial role in various modern business sectors, including climate monitoring [22, 13, 14], healthcare [12, 3, 46], urban management [43, 16, 15], and online service [35]. Time series are derived from diverse sources with inherent characteristics, such as system configurations, alongside external influential factors like environmental status. These factors are considered *attributes* of the time series. In many real-world scenarios, such as healthcare and urban monitoring, time series data tend to be sparse and privacy-sensitive, particularly concerning specific attributes that are rarely observed in real-world scenarios.

Synthesizing time series data has emerged as a prominent research area aimed at addressing these challenges. Existing methods for time series synthesis range from unconditional generation [47, 31] to conditional generation [39, 27, 4]. Unconditional generation produces outputs based solely on the underlying distribution of the data. Without relying on any input conditions, the generated results are highly uncontrollable. Conditional generation controls the outcomes of the generated data based on the input conditions. However, conditional generation tends to produce samples around the data mean [40], as demonstrated in our experiments, which could easily overlook detailed characteristics of the data, such as high-frequency patterns and local structures. Both unconditional and conditional generation paradigms synthesize time series from scratch, and they are incapable of manipulating existing samples. As a result, these paradigms are unable to answer *"what if"* questions in time series synthesis: *given a time series, what would it become if some of its attributes are modified?*

---

[*]Baoyu Jing and Shuqi Gu share the co-first authorship.
[†]Correspondence to Kan Ren: renkan@shanghaitech.edu.cn

38th Conference on Neural Information Processing Systems (NeurIPS 2024).

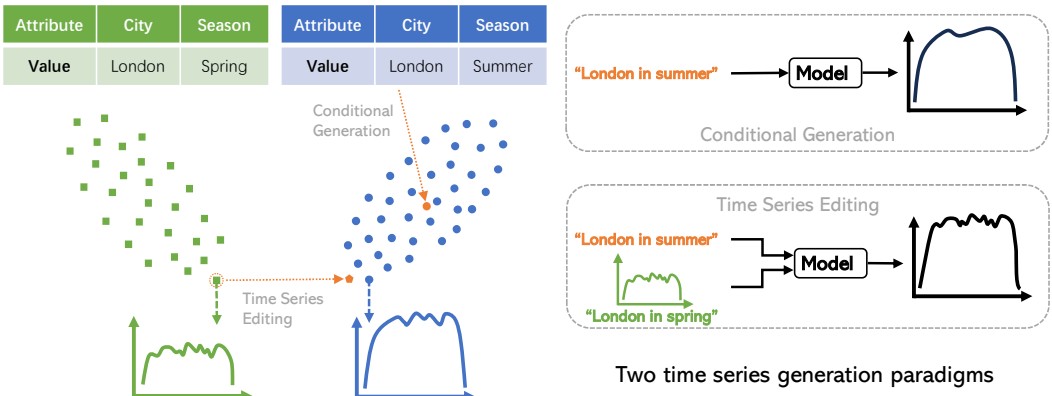

Figure 1: Illustration of the time series generation paradigms. Conditional generation generates time series from scratch, which usually generates samples around the dataset mean. Time series editing allows for the manipulation of the attributes of an input time series sample, which aligns with the desired target attribute values while preserving other properties.

In this paper, we introduce a novel task, called **Time Series Editing (TSE)**, for sample-level time series manipulation. More specifically, our objective is to directly manipulate specific attributes of an input time series to target values while maintaining consistency in other characteristics. For instance, consider observing air quality under season conditions in different cities as shown in Fig. 1. On the left side, two example air quality time series for the spring and summer of London are illustrated. The spring one exhibits a lower amplitude with more noise, whereas the summer one features a higher amplitude with less noise. What would the resulting time series look like if we modify the season attribute of the spring one to summer while maintaining all other properties? As illustrated in Fig. 1, it could have a larger amplitude with much noise.

The task of TSE is complex due to several challenges. Firstly, the time series data distribution over the full composited attribute space is biased and may not be adequately covered, leaving gaps in our understanding, especially concerning unobservable or poorly defined attributes. For example, in climate data analysis, attributes like temperature and humidity are observable and well-defined. However, attributes like atmospheric pressure variations or localized microclimates may be challenging to observe or define accurately. Secondly, different attributes influence time series at varying resolutions. For example, trends have a global impact, while seasonality exerts a more localized influence. Modeling these multi-scale attributes and time series associations, while effectively controlling them, presents significant difficulties.

To address these challenges, we introduce a novel method called **Time Series Editor (TEdit)**, which is based on predominant generative models, specifically diffusion models [10, 38, 39, 27]. To address the data coverage issue, we propose a novel bootstrap learning algorithm, which leverages the generated data as pseudo-supervision for subsequent model learning. This algorithm helps improve the coverage of the whole attribute space and enhance the generation performance. To capture the intricate multi-scale associations between time series and attributes, we introduce a multi-resolution modeling and generation paradigm. The proposed multi-resolution paradigm can manipulate the given time series and attributes both effectively and efficiently. Our experiments, conducted on both synthetic and real-world datasets, demonstrate the effectiveness of our proposed solution. Specifically, our approach excels in generating precise time series under specified attributes, while keeping consistency in other attributes, showcasing the practical utility of our method.

In summary, our main contributions are threefold. (1) We introduce and formulate the novel task of Time Series Editing (TSE). To the best of our knowledge, this is the first work exploring editing time series according to the given attribute configuration upon the input data. (2) We propose a novel multi-resolution diffusion model, called TEdit, with a bootstrap learning algorithm, which can flexibly capture the patterns at diverse granularity and gradually improve the generation performance. (3) We conduct and release a benchmark for TSE, including synthetic data and several carefully curated real-world datasets with a comprehensive evaluation protocol, aiming to facilitate further research in the community.

## 2 Background

### 2.1 Time Series Generation

A time series sample $\mathbf{x} \in \mathbb{R}^L$ is a series of $L$ chronologically ordered sample points. Each time series sample adheres to some attributes $\mathbf{a} \in \mathbb{N}_+^K$, such as trend and the cycle number, and the $k$-th item $a_k$ corresponds to $N_k$ categorical options where $k \in \{1, ..., K\}$. In real-world scenarios, each time series may have several attributes, e.g., the air quality time series is associated with $K = 2$ attributes: location and season. Note that, not all the attributes can be observed due to practical limitations.

Before describing the main task of the paper, we first briefly review the background of Time Series Generation. *Unconditional Time Series Generation (UTSG)* samples the time series data $\mathbf{x}$ upon the modeled data distribution $p(\mathbf{x})$ such that $\mathbf{x} \sim p(\mathbf{x})$, of which the generation process is uncontrollable. *Conditional Time Series Generation (CTSG)* generates the time series sample $\mathbf{x}$ based on the conditional distribution such that $\mathbf{x} \sim p(\mathbf{x}|\mathbf{a})$, where $\mathbf{a}$ is the input condition. CTSG tends to produce data samples near the dataset mean [40]. Both UTSG and CTSG paradigms generate time series from scratch and lack the ability to directly manipulate the existing samples.

### 2.2 Conditional Diffusion Models

Diffusion models [10, 37, 38] learn to estimate and remove the random noise, which was added in the forward process onto the real-world data, through a sequence of sampling processes. Specifically, during training, random noise is gradually added to the original data sample $\mathbf{x}_0 = \mathbf{x}^1$ via a Gaussian Markov transition $q(\mathbf{x}_t|\mathbf{x}_{t-1}) := \mathcal{N}(\sqrt{1-\beta_t}\mathbf{x}_{t-1}, \beta_t\mathbf{I})$, where $t \in [1, T]$ indicates the diffusion step, and $\{\beta_t\}_{t=0}^T$ are the predetermined variance schedule. The expression of latent variable $\mathbf{x}_t$ can be simplified as $\mathbf{x}_t = \sqrt{\alpha_t}\mathbf{x}_0 + \sqrt{1-\alpha_t}\boldsymbol{\epsilon}$, $\boldsymbol{\epsilon} \sim \mathcal{N}(0, \mathbf{I})$, $\alpha_t := \Pi_{s=1}^t(1-\beta_s)$. The learnable component of the diffusion models is a noise estimation network $\boldsymbol{\epsilon}_\theta(\mathbf{x}_t, t, \mathbf{a})$, where $\mathbf{a}$ denotes the additional condition such as attributes. $\boldsymbol{\epsilon}_\theta(\mathbf{x}_t, t, \mathbf{a})$ is trained by estimating the noise added to $\mathbf{x}_t$:

$$\min_\theta \mathcal{L}(\mathbf{x}_0) := \min_\theta \mathbb{E}_{\boldsymbol{\epsilon} \sim \mathcal{N}(\mathbf{0},\mathbf{I}), t \sim \mathcal{U}(1,T)} \|\boldsymbol{\epsilon} - \boldsymbol{\epsilon}_\theta(\mathbf{x}_t, t, \mathbf{a})\|_2^2 \tag{1}$$

where $\mathbf{x}_0 \sim q(\mathbf{x}_0)$ is sampled from the real data distribution, $\mathbf{x}_t$ is a noisy version of $\mathbf{x}_0$.

After training, given an attribute $\mathbf{a}$, we can generate a sample $\hat{\mathbf{x}}_0$ from random noise $\hat{\mathbf{x}}_T \sim \mathcal{N}(0, \mathbf{I})$ using $\boldsymbol{\epsilon}_\theta(\mathbf{x}_t, t, \mathbf{a})$ with a sampler, e.g., deterministic Denoising Diffusion Implicit Model (DDIM) [38]:

$$\hat{\mathbf{x}}_{t-1} = \sqrt{\alpha_{t-1}}\mathbf{f}_\theta(\hat{\mathbf{x}}_t, t, \mathbf{a}) + \sqrt{1-\alpha_{t-1}}\boldsymbol{\epsilon}_\theta(\hat{\mathbf{x}}_t, t, \mathbf{a}) \tag{2}$$

$$\mathbf{f}_\theta(\hat{\mathbf{x}}_t, t, \mathbf{a}) = (\hat{\mathbf{x}}_t - \sqrt{1-\alpha_t}\boldsymbol{\epsilon}_\theta(\hat{\mathbf{x}}_t, t, \mathbf{a}))/\sqrt{\alpha_t} \tag{3}$$

where $\mathbf{f}_\theta(\hat{\mathbf{x}}_t, t, \mathbf{a})$ can be regarded as a prediction of $\mathbf{x}_0$ at the diffusion step $t$.

Adapting diffusion methods from the computer vision domain to the time series domain is not trivial. Time series is different from images, and it poses unique challenges for modeling and generating time series data, such as the complex multi-scale entanglement between time series and attributes.

## 3 Time Series Editing

Given a time series and its associated attributes $(\mathbf{x}, \mathbf{a})$, a natural question arises: *"what would it become if some of its attributes are modified?"* In our work, we take a novel perspective of directly editing the given time series with the specified attribute modifications.

In this section, we first formally formulate the problem of *Time Series Editing (TSE)* in Sec. 3.1. Next, we present our method *Time Series Editor (TEdit)* with the overall procedure of the diffusion model-based TSE in Sec. 3.2, the model architecture in Sec. 3.3 and the learning algorithm in Sec. 3.4.

### 3.1 Problem Formulation

Recall that, each time series sample $\mathbf{x} \in \mathbb{R}^L$ is associated with a set of attributes $\mathbf{a} \in \mathbb{N}_+^K$. Each attribute $a_k$ has $N_k$ options, $k \in \{1, ..., K\}$. For example, the trend type may contain $N_k = 4$ values

---

[1]$\mathbf{x}_0$ and $\mathbf{x}$ are interchangeable in this paper.

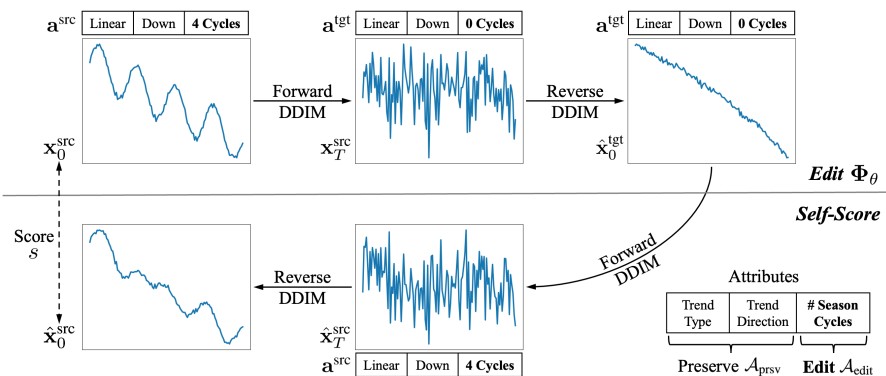

Figure 2: Illustration of the editing function $\Phi_\theta$ (upper) and the process of self-scoring the generated $\hat{\mathbf{x}}_0^{\text{tgt}}$ for bootstrap learning (lower). **Upper**: $\Phi_\theta(\mathbf{x}_0^{\text{src}}, \mathbf{a}^{\text{src}}, \mathbf{a}^{\text{tgt}})$ first encodes the source $(\mathbf{x}_0^{\text{src}}, \mathbf{a}^{\text{src}})$ into the latent viable $\mathbf{x}_T^{\text{src}}$ via the forward DDIM Eq.(4), and then decodes $\mathbf{x}_T^{\text{src}}$ with the target attributes $\mathbf{a}^{\text{tgt}}$: $(\mathbf{x}_T^{\text{src}}, \mathbf{a}^{\text{tgt}})$ into $\hat{\mathbf{x}}_0^{\text{tgt}}$ via the reverse DDIM in Eq.(5). See Sec. 3.2 for more details. **Lower**: during bootstrap learning, we use $\Phi_\theta$ to self-score the generated $\hat{\mathbf{x}}_0^{\text{tgt}}$ by editing $\hat{\mathbf{x}}_0^{\text{tgt}}$ back to $\hat{\mathbf{x}}_0^{\text{src}}$, and obtain the score $s = \text{MSE}(\hat{\mathbf{x}}_0^{\text{src}}, \mathbf{x}_0^{\text{src}})$, see Sec. 3.4 for more details.

like linear, quadratic, exponential, and logistic. Let $(\mathbf{x}^{\text{src}}, \mathbf{a}^{\text{src}})$ be a pair of *source time series* and *source attributes* to be edited, and $\mathbf{a}^{\text{tgt}}$ be the desired *target attributes*. Comparing $\mathbf{a}^{\text{tgt}}$ and $\mathbf{a}^{\text{src}}$, there are $K_{\text{edit}}$ edited attributes, e.g., trend types, and $K_{\text{prsv}}$ preserved attributes, e.g., cycle numbers, where $K_{\text{edit}} + K_{\text{prsv}} = K$. Now, we can formally define the task as below.

**Definition 1 (Time Series Editing (TSE))** *The time series editing task is to build a function $\Phi_\theta$ to generate a target time series $\hat{\mathbf{x}}^{\text{tgt}} = \Phi_\theta(\mathbf{x}^{\text{src}}, \mathbf{a}^{\text{src}}, \mathbf{a}^{\text{tgt}})$ by modifying the set of $K_{\text{edit}}$ edited attributes $\mathcal{A}_{\text{edit}}$ and maintaining the set of $K_{\text{prsv}}$ preserved attributes $\mathcal{A}_{\text{prsv}}$ as well as other information of $\mathbf{x}^{\text{src}}$.*

### 3.2 Editing with Source Modeling and Target Generation

Given a pair of source time series and attributes $(\mathbf{x}^{\text{src}}, \mathbf{a}^{\text{src}})$, generating the target time series $\hat{\mathbf{x}}^{\text{tgt}}$ corresponding to the target attributes $\mathbf{a}^{\text{tgt}}$ entails two requirements. On one hand, the edited attributes $\mathcal{A}_{edit}$ should be satisfied after the generation. On the other hand, the preserved attributes $\mathcal{A}_{prsv}$ as well as other characteristics, e.g., noise, need to be maintained. Conditional generation [27], such as directly adopting the conditional diffusion model described in Sec. 2.2, would fail the editing task because it does not take into consideration the detail characteristics of the source time series.

In this paper, we propose a diffusion based two-stage procedure for TSE: $\Phi_\theta(\mathbf{x}^{\text{src}}, \mathbf{a}^{\text{src}}, \mathbf{a}^{\text{tgt}})$, which is illustrated in the upper part of Fig. 2. In the first stage, we encode both the attribute semantics [6] and the detail characteristics [10] of the source time series $\mathbf{x}_0^{\text{src}}$ into the latent variable $\mathbf{x}_T^{\text{src}}$ via the deterministic forward DDIM process [18]:

$$\mathbf{x}_{t+1}^{\text{src}} = \sqrt{\alpha_{t+1}}\mathbf{f}_\theta(\mathbf{x}_t^{\text{src}}, t, \mathbf{a}^{\text{src}}) + \sqrt{1 - \alpha_{t+1}}\boldsymbol{\epsilon}_\theta(\mathbf{x}_t^{\text{src}}, t, \mathbf{a}^{\text{src}}). \tag{4}$$

In the second stage, originating from the latent variable $\hat{\mathbf{x}}_T^{\text{tgt}} = \mathbf{x}_T^{\text{src}}$, we gradually generate the final target time series $\hat{\mathbf{x}}_0^{\text{tgt}}$ with the consideration of the target attribute $\mathbf{a}^{\text{tgt}}$ via the deterministic reverse DDIM process [38]:

$$\hat{\mathbf{x}}_{t-1}^{\text{tgt}} = \sqrt{\alpha_{t-1}}\mathbf{f}_\theta(\hat{\mathbf{x}}_t^{\text{tgt}}, t, \mathbf{a}^{\text{tgt}}) + \sqrt{1 - \alpha_{t-1}}\boldsymbol{\epsilon}_\theta(\hat{\mathbf{x}}_t^{\text{tgt}}, t, \mathbf{a}^{\text{tgt}}), \tag{5}$$

where $\mathbf{f}_\theta$ is given in Eq. (3). Till now, we have detailed the procedure of the proposed TEdit. In the following subsections, we introduce the model architecture and the training algorithm.

### 3.3 Multi-Resolution Noise Estimator

The key component of our TEdit is the noise estimator $\boldsymbol{\epsilon}_\theta(\mathbf{x}, t, \mathbf{a})$ in Eqs. (4)(5). Though many diffusion model realizations have been proposed in other fields [18, 10, 37, 38, 24] and the time series domain [39, 27], we found them inefficacious in modeling and generating time series data, especially

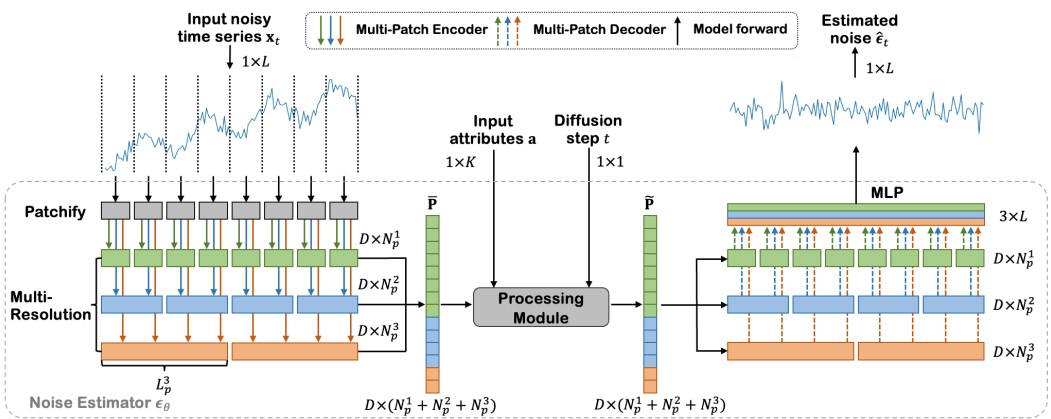

Figure 3: Architecture of the proposed multi-resolution noise estimator $\epsilon_\theta$. We illustrate with $R = 3$ patching schema, patch length $L_p^r = 2^{r-1}, r \in \{1, ..., R\}$ and the input length $L = 8$. $N_p^r = \lfloor \frac{L - L_p^r}{L_p^r} \rfloor$ is the patch number. $D$ is the embedding size. Please refer to Sec. 3.3 for details.

when considering attributes. Few of them consider that different attributes can influence the time series on varying scales. We argue that it is important to take into account these differences. For instance, the trend types have a global impact on time series, while the seasonality affects the time series in local areas.

In this paper, we propose a multi-resolution noise estimator. It first slices the input time series $\mathbf{x}_t$ into several patch sequences of different resolutions $r \in \{1, ..., R\}$, where $R$ is the total number of resolutions. Then it processes the patch sequences along with other information, e.g., attributes $\mathbf{a}$ and diffusion step $t$, to estimate the noise $\hat{\epsilon}_t^r$ for resolution $r$. Finally, the estimated noise for different resolutions are mixed together to obtain the final estimated noise $\hat{\epsilon}_t$. An illustration is presented in Fig. 3. We elaborate on the details in the following content[2].

**Patchifying.** Following [29], we patchify the input time series $\mathbf{x} \in \mathbb{R}^L$, into a sequence of patch tokens $\mathbf{P} = \{\mathbf{p}_1, ..., \mathbf{p}_{N_p}\}$, where $\mathbf{p}_i \in \mathbb{R}^{L_p}$, $L_p$ is the window size and $N_p = \lfloor \frac{L - L_p}{L_p} \rfloor$ is the patch number. After that, we encode them into embeddings $\bar{\mathbf{P}} = \{\bar{\mathbf{p}}_1, ..., \bar{\mathbf{p}}_{N_p}\}$, where $\bar{\mathbf{p}}_i \in \mathbb{R}^D$.

**Multi-Resolution Modeling and Generation.** To model multi-resolution patterns and better control the conditional generation at multiple scales, we propose a multi-patch design with various patch lengths. Specifically, as shown in the left part of Fig. 3, following the above patchifying operation, we encode the input time series $\mathbf{x}$ into patch embedding sequences $\{\bar{\mathbf{P}}^r\}_{r=1}^R$ of $R$ resolutions, where $\bar{\mathbf{P}}^r = \{\bar{\mathbf{p}}_1^r, ..., \bar{\mathbf{p}}_{N_p^r}^r\}$, $\bar{\mathbf{p}}_i^r \in \mathbb{R}^D$, $N_p^r = \lfloor \frac{L - L_p^r}{L_p^r} \rfloor$ is the patch number, $L_p^r$ is the window size. We set the patch length as $L_p^r = b^{r-1}$ to produce exponential receptions, where $b \in \mathbb{N}_+$ is the base. Thereafter, the processing module operates self-attention as the Transformer [42] to capture the input patterns for each resolution $r$, and it incorporates the attribute $\mathbf{a}$ and diffusion step $t$ information to produce the embeddings $\{\tilde{\mathbf{P}}^r\}_{r=1}^R$. The details of the processing module are presented in Appendix D.

The processed output $\{\tilde{\mathbf{P}}^r\}_{r=1}^R$ is still in the form of multi-resolution sequences. The model subsequently decodes the diffusion noise at different resolutions back into the original space and integrates the outcomes at different resolutions to produce the final noise estimation, as shown on the right side of Fig. 3. Specifically, for each $\tilde{\mathbf{P}}^r$, the decoder transforms it to $\hat{\epsilon}^r \in \mathbb{R}^L$ in the original time series space. The final estimated noise is obtained via a mixing upon the concatenation of estimated noise of different resolutions as $\hat{\epsilon} = \text{MLP}([\epsilon^1, ..., \epsilon^R])$, where $\text{MLP}(\cdot)$ denotes Multi-Layer Perceptron.

**Parallel Processing.** Though more effective, the proposed multi-resolution modeling and generation brings efficiency issues since it derives multiple token sequences with various sequence lengths. Iteratively processing each sequence is of low efficiency. Herein we design a novel parallelization

---

[2]We discuss the noise estimator in the diffusion step $t$ here, thus, we omit the notation of the diffusion step $t$ without causing confusion.

with an attention masking mechanism in the processing module in Fig. 3 to harness the parallelism of the Graphics Processing Unit (GPU). Specifically, we first concatenate the input patch embedding sequences of different resolutions into a single vector $\bar{\mathbf{P}} = [\bar{\mathbf{P}}^1, ..., \bar{\mathbf{P}}^R] \in \mathbb{R}^{D \times (N_p^1 + ... + N_p^R)}$. When calculating pair-wise self-attention scores as in the Transformer [42], we use a mask matrix to mask out the inter-sequence attention operations across different sequences while preserving intra-sequence attention. Please refer to Appendix D for more details.

### 3.4 Bootstrap Learning Algorithm

Although any pretrained conditional diffusion model $\epsilon_\theta$ can be directly used to perform editing as shown in Sec. 3.2, the pretrained model is an essentially conditional generator, which has a great ability in generating time series directly from attributes but it might be less effective in modifying the attributes of an existing time series. The simplest way to improve the model's ability is to finetune $\epsilon_\theta$ via the ground-truth source $(\mathbf{x}_0^{\text{src}}, \mathbf{a}^{\text{src}})$ and target $(\mathbf{x}_0^{\text{tgt}}, \mathbf{a}^{\text{tgt}})$ pairs. However, the imaginary target $\mathbf{x}_0^{\text{tgt}}$, which satisfies both $\mathbf{a}^{\text{tgt}}$ and the details of $\mathbf{x}_0^{\text{src}}$ might be very rare or even does not exist in the real world. Thus, a key challenge for finetuning is how to effectively learn the information of $\mathbf{x}_0^{\text{tgt}}$.

Fortunately, the pretrained model is capable of generating some edited samples of a certain quality via $\hat{\mathbf{x}}_0^{\text{tgt}} = \Phi_\theta(\mathbf{x}_0^{\text{src}}, \mathbf{a}^{\text{src}}, \mathbf{a}^{\text{tgt}})$. An example is shown in Fig. 2. In this paper, we propose a bootstrap learning algorithm, which first *pretrains* $\epsilon_\theta$ based on the noise estimation loss $\mathcal{L}$ in Eq. (1) and then *finetunes* $\epsilon_\theta$ based on $\hat{\mathbf{x}}_0^{\text{tgt}}$ with top confidence scores. Here, we briefly present the key steps of the bootstrap learning, and the full algorithm is given in Appendix E. Given a batch $(\mathbf{X}_0^{\text{src}}, \mathbf{A}^{\text{src}})$, where $\mathbf{X}_0^{\text{src}} \in \mathbb{R}^{B \times L}$, $\mathbf{A}^{\text{src}} \in \mathbb{N}_+^{B \times K}$, $B$ is the batch size, the finetuning works as follows:

- **Compose target attributes $\mathbf{A}^{\text{tgt}}$.** For each $(\mathbf{x}_0^{\text{src}}, \mathbf{a}^{\text{src}}) \in (\mathbf{X}_0^{\text{src}}, \mathbf{A}^{\text{src}})$, we compose the imaginary target attributes $\mathbf{a}^{\text{tgt}}$ based on $\mathbf{a}^{\text{src}}$ by randomly sampling values for edited attributes $\mathcal{A}_{\text{edit}}$ and keeping the values of the preserved attributes $\mathcal{A}_{\text{prsv}}$. Then we have the tuple $(\mathbf{X}_0^{\text{src}}, \mathbf{A}^{\text{src}}, \mathbf{A}^{\text{tgt}})$.

- **Generate the edited time series $\hat{\mathbf{X}}_0^{\text{tgt}}$ via $\Phi_\theta$.** An illustration is shown in the upper part of Fig. 2.

- **Self-score $\hat{\mathbf{X}}_0^{\text{tgt}}$ via $\Phi_\theta$ and keep top $\psi$ samples.** An illustration of the self-scoring process is shown in the lower part of Fig. 2. We first edit $\hat{\mathbf{X}}_0^{\text{tgt}}$ back to source $\hat{\mathbf{X}}_0^{\text{src}} = \Phi_\theta(\hat{\mathbf{X}}_0^{\text{tgt}}, \mathbf{A}^{\text{tgt}}, \mathbf{A}^{\text{src}})$. Then we use the Mean Squared Error (MSE) between $\mathbf{X}_0^{\text{src}}$ and $\hat{\mathbf{X}}_0^{\text{src}}$ to score $\hat{\mathbf{X}}_0^{\text{tgt}}$. The top $\psi$ samples of $\hat{\mathbf{X}}_0^{\text{tgt}}$ with the lowest MSE are selected as the bootstrap samples, denoted by $\hat{\mathbf{X}}_{0,\text{bs}}^{\text{tgt}}$.

- **Update $\epsilon_\theta$ by minimizing $\mathcal{L}_{\textbf{BS}}$.** We finetune $\epsilon_\theta$ by minimizing the noise estimation loss $\mathcal{L}$ in Eq. (1) for $\hat{\mathbf{X}}_{0,\text{bs}}^{\text{tgt}}$. Formally, we minimize $\mathcal{L}_{\text{BS}} = \mathcal{L}(\hat{\mathbf{X}}_{0,\text{bs}}^{\text{tgt}})$.

## 4 Experiments

In this section, we present experiments aimed at addressing the following research questions: **RQ1**: How does the proposed TEdit perform in terms of editing and preserving attributes? **RQ2**: What's the impact of bootstrap training? **RQ3**: What's the impact of multi-resolution modeling?

### 4.1 Experimental Setup

**Datasets.** We collect three datasets for TSE, including one synthetic dataset and two real-world datasets. For the **Synthetic** data, each time series sample has a length of 128, and is associated with 3 attributes, where there are 4 trend types, 2 trend directions, and 4 season cycles. In addition to attributes, noise and bias are added to simulate real-world conditions. The **Air Quality** [8] dataset contains PM2.5 time series of Beijing and London from 01/01/2017 to 31/03/2018. Each time series has a length of 168 and is affected by two attributes: 2 cities and 4 seasons. The **Motor Imagery** [1] dataset contains the ElectroEncephaloGram (EEG) data of subjects, who were required to imagine the movements of either the tongue or the left small finger. Each sample takes a length of 150 with 2 attributes: 2 movements and 64 channel ids.

Each of the three datasets is associated with a pertaining dataset and several finetuning datasets. The pretraining datasets only contain the source time series $\mathbf{x}^{\text{src}}$ and its attributes $\mathbf{a}^{\text{src}}$. During pertaining, the noise estimator $\epsilon_\theta$ is trained to denoise $\mathbf{x}_t^{\text{src}}$ with $\mathbf{a}^{\text{src}}$ via Eq. (1). Each finetuning

dataset corresponds to a specific split of $\mathcal{A}_{\text{edit}}$ and $\mathcal{A}_{\text{prsv}}$. For example, the Synthetic dataset has 3 attributes, and thus it has $6 = \sum_{i=1}^{2} \binom{3}{i}$ different splits, namely, finetuning datasets. Similarly, both Air Quality and Motor Imagery datasets have 2 finetuning datasets. Finetuning datasets contain the source time series $\mathbf{x}^{\text{src}}$, source attributes $\mathbf{a}^{\text{src}}$, and target attributes $\mathbf{a}^{\text{tgt}}$. The process of generating the target attributes mirrors the target attribute composition step in bootstrap learning. Note that for Synthetic, the target time series $\mathbf{x}^{\text{tgt}}$ is also available. During finetuning, we use the proposed bootstrap learning algorithm to further improve the model. More details can be found in Appendix F.

**Evaluation Metrics.** A good editor should be able to generate $\hat{\mathbf{x}}^{\text{tgt}}$ that aligns with each $a_k^{\text{tgt}} \in \mathbf{a}^{\text{tgt}}$. The cornerstone of our evaluation is the **Contrastive Time series Attribute Pretraing (CTAP)** model, which extracts time series and attribute embeddings by learning their alignment. Our CTAP is similar to CLIP [33]. For details of CTAP, please refer to Appendix G. Based on the CTAP model, we introduce two metrics: **CTAP score** and **Log Ratio of Target-to-Source (RaTS)**. (1) The CTAP score is similar to the CLIP-I score [23, 33], which measures the alignment between the generated time series and the real-world time series which are associated with the given attribute value $a_k^{\text{tgt}}$. Specifically, we first use the CTAP model to extract embeddings $\hat{\mathbf{h}}_{\mathbf{x}}$, which is the embedding of $\hat{\mathbf{x}}^{\text{tgt}}$, and $\bar{\mathbf{h}}_{\mathbf{x}}|a_k^{\text{tgt}}$, which is the average embedding of the time series associated with $a_k^{\text{tgt}}$ in the training data. Then we calculate the cosine similarity of $\hat{\mathbf{h}}_{\mathbf{x}}$ and $\bar{\mathbf{h}}_{\mathbf{x}}|a_k^{\text{tgt}}$, where the higher similarity indicates better alignment. (2) The RaTS score measures whether $\hat{\mathbf{x}}^{\text{tgt}}$ is closer to $a_k^{\text{tgt}}$ than $\mathbf{x}^{\text{src}}$. Formally, the RaTS score for a tuple $(\hat{\mathbf{x}}^{\text{tgt}}, \mathbf{x}^{\text{src}}, a_k^{\text{tgt}})$ is defined as:

$$\text{RaTS}(\hat{\mathbf{x}}^{\text{tgt}}, \mathbf{x}^{\text{src}}, a_k^{\text{tgt}}) = \log\left(\frac{p(a_k^{\text{tgt}}|\hat{\mathbf{x}}^{\text{tgt}})}{p(a_k^{\text{tgt}}|\mathbf{x}^{\text{src}})}\right), \tag{6}$$

where $p(a_k|\mathbf{x})$ is calculated by applying a softmax over the similarity scores of all $(\mathbf{h}_{\mathbf{x}}, \mathbf{h}_{a_k=i})$ pairs, where $\mathbf{h}_{\mathbf{x}}$ is the CTAP embedding of $\mathbf{x}$, and $\mathbf{h}_{a_k=i}$, $i \in \{1, ..., N_k\}$, is the CTAP embedding of the $i$-th possible value for the $k$-th attribute $a_k$.

We mainly evaluate $\hat{\mathbf{x}}^{\text{tgt}}$ from two perspectives: *editability* and *preservability* for edited attributes $\mathcal{A}_{\text{edit}}$ and preserved attributes $\mathcal{A}_{\text{prsv}}$, respectively. For the edited attributes $\mathcal{A}_{\text{edit}}$, the higher RaTS and CTAP the better. For the preserved attributes $\mathcal{A}_{\text{prsv}}$, the higher CTAP the better. Since RaTS can take negative values, we use |RaTS| for $\mathcal{A}_{\text{prsv}}$, which measures the semantic divergence of $\hat{\mathbf{x}}^{\text{tgt}}$ from $\mathbf{x}^{\text{src}}$ w.r.t $a_k$. The lower |RaTS| means the better preservability. For more details, please refer to Appendix H. In addition to the semantic level evaluations, for the synthetic data, since we have ground truth target time series, we also use Mean Square Error (MSE), Mean Absolute Error (MAE) to perform the point level evaluation.

**Compared Methods.** Since there is no existing method for TSE, we modify the popular time series diffusion model CSDI [39] and the recent conditional diffusion model Time Weaver [27] for TSE. As CSDI is unable to process attributes, we add an extra attribute encoder to incorporate attribute information. For Time Weaver, we slightly modify its attribute encoder for our settings. After pretraining the modified CSDI and Time Weaver, we use them to edit time series via the editing procedure described in Sec. 3.2. Our proposed TEdit-CSDI and TEdit-TW are implemented by incorporating core processing modules of CSDI and Time Weaver into our proposed multi-resolution noise estimator Sec. 3.3, and trained via Sec. 3.4. Architecture details are presented in Appendix D.

**Implementation Details.** For all experiments, we set the number of diffusion steps as $T = 50$, embedding size for attributes and time series as 64, and use Adam optimizer [20] to train the model. For pretraining, we set (batch size, learning rate) as (256, 1e-3); for finetuning, we set them as (64,1e-7) for Synethtic and (32,1e-7) for Air Quality and Motor Imagery. We conduct a grid search for the hyperparameters of the multi-resolution. Considering the balance between performance and efficiency, we chose a compromise ratio for bootstrap. Specifically, $(R, L_p, \psi) = (3, 2, 0.5)$ for Synthetic, $(3, 2, 0.5)$ for Air Quality, $(3, 3, 0.5)$ for Motor Imagery. All our experiments were conducted on a single Nvidia-A100 GPU.

## 4.2 Main Results

In this section, we quantitatively evaluate the performance of comparison methods for editing and preserving attributes (**RQ1**) for all the datasets. For the edited attributes $\mathcal{A}_{\text{edit}}$, we report the RaTS and CTAP scores to depict the models' ability to edit attributes. For the preserved attributes $\mathcal{A}_{\text{prsv}}$,

| | Synthetic | | | | | | Air | | | | Motor | | | |
|---|---|---|---|---|---|---|---|---|---|---|---|---|---|---|
| | Overall | | Edited | | Preserved | | Edited | | Preserved | | Edited | | Preserved | |
| | ↓MSE | ↓MAE | ↑RaTS | ↑CTAP | ↓\|RaTS\| | ↑CTAP | ↑RaTS | ↑CTAP | ↓\|RaTS\| | ↑CTAP | ↑RaTS | ↑CTAP | ↓\|RaTS\| | ↑CTAP |
| CSDI | 0.1789 | 0.3221 | 0.7540 | 0.5405 | 0.1439 | 0.7898 | 0.7452 | 0.1581 | 0.1705 | 0.6311 | 0.0939 | 0.4203 | 0.1597 | 0.6617 |
| Time Weaver | 0.1454 | 0.2898 | 0.9030 | 0.6943 | 0.1169 | 0.8292 | 0.8956 | 0.3266 | 0.1866 | 0.6299 | 0.0979 | 0.4168 | **0.1520** | **0.6691** |
| TEdit-CSDI | **0.1235** | **0.2606** | 0.9257 | 0.7109 | 0.1021 | 0.8553 | 0.8022 | 0.2179 | **0.1614** | **0.6529** | 0.1016 | 0.4186 | 0.1580 | 0.6654 |
| TEdit-TW | 0.1315 | 0.2722 | **1.0121** | **0.7957** | **0.0995** | **0.8622** | **0.9661** | **0.3930** | 0.1916 | 0.6274 | **0.1212** | **0.4348** | 0.1571 | 0.6621 |

Table 1: Averaged performance over all finetuning sets for Synthetic (left), Air (middle), and Motor (right). "Edited" and "Preserved" are the average results of all edited and preserved attributes.

| | Synthetic | | | | | | Air | | | | Motor | | | |
|---|---|---|---|---|---|---|---|---|---|---|---|---|---|---|
| | Overall | | Edited | | Preserved | | Edited | | Preserved | | Edited | | Preserved | |
| | ↓MSE | ↓MAE | ↑RaTS | ↑CTAP | ↓\|RaTS\| | ↑CTAP | ↑RaTS | ↑CTAP | ↓\|RaTS\| | ↑CTAP | ↑RaTS | ↑CTAP | ↓\|RaTS\| | ↑CTAP |
| TEdit-TW w GT | **0.1233** | **0.2622** | **1.0165** | **0.7991** | 0.0984 | **0.8650** | - | - | - | - | - | - | - | - |
| TEdit-TW | 0.1315 | 0.2722 | 1.0121 | 0.7957 | 0.0995 | 0.8622 | **0.9661** | **0.3930** | 0.1916 | 0.6274 | **0.1212** | **0.4348** | 0.1571 | 0.6621 |
| w/o BS | 0.1376 | 0.2793 | 1.0127 | 0.7952 | **0.0962** | 0.8632 | 0.9524 | 0.3792 | **0.1839** | **0.6418** | 0.1113 | 0.4289 | 0.1571 | 0.6658 |
| w/o BS & MR | 0.1454 | 0.2898 | 0.9030 | 0.6943 | 0.1169 | 0.8292 | 0.8956 | 0.3266 | 0.1866 | 0.6299 | 0.0978 | 0.4168 | **0.1520** | **0.6691** |

Table 2: Ablation studies on the Synthetic, Air and Motor datasets. GT, BS and MR refer to Ground Truth source and target pairs, BootStrap and Multi-Resolution. Results are averaged over all finetuning sets. "Edited" and "Preserved" are the average results of all edited and preserved attributes.

we report the |RaTS| and CTAP scores to depict the models' ability to preserve attributes. For the Synthetic dataset, since we have the ground truth, we also include MSE and MAE to evaluate the overall disparity between the ground truth $x^{tgt}$ and the generated $\hat{x}^{tgt}$.

In Tab. 1, we present the averaged results over all the finetuning sets for each dataset. The detailed results for each finetuning sets are presented in Appendix I.1. Firstly, for the overall performance (MSE and MAE), the proposed TEdit could significantly outperform baselines. Secondly, for the edited attributes $\mathcal{A}_{edit}$, TEdit-CSDI and TEdit-TW could respectively outperform CSDI and Time Weaver on RaTS and TAP, showing that $\hat{x}^{tgt}$ generated by TEdit could better fulfill the desired target $\mathcal{A}_{edit}$. Thirdly, for the preserved attributes $\mathcal{A}_{prsv}$, TEdit could basically maintain |RaTS| and TAP scores, which indicates TEdit is capable of preserving $\mathcal{A}_{prsv}$. In summary, these observations show that TEdit, including the multi-resolution and bootstrap learning, could improve the ability of conditional diffusion models to edit $\mathcal{A}_{edit}$ and preserve $\mathcal{A}_{prsv}$.

### 4.3 Ablation Study

In this subsection, we conduct ablation studies on the Synthetic and Motor datasets based on the TEdit-TW to investigate the impact of the proposed multi-resolution modeling (**RQ2**) and the bootstrap training algorithm (**RQ3**). The averaged results over all the finetuning sets are presented in Tab. 2.

Firstly, we compare TEdit-TW trained with BS (BootStrap) with TEdit-TW trained with GT (Ground Truth) on the Synthetic dataset[3]. For GT, we finetune TEdit-TW by minimizing MSE between the generated $\hat{x}^{tgt}$ and the ground truth $x^{tgt}$. It can be observed that GT performs much better than BS on the overall metrics (MSE and MAE), and it is only slightly better than BS on the attribute level metrics for both edited and preserved attributes, demonstrating that BS has a strong ability to capture the attribute semantic information. Secondly, we remove BS from TEdit-TW and compare its performance with the full TEdit-TW model. For the Synthetic dataset, BS could improve the overall scores, and maintain the performance on other attribute level metrics. For the Air dataset, BS primarily improves the scores of the edited attributes, with a trade-off in the scores of the preserved attributes. We hypothesize that there might exist some intrinsic connections between the two attributes: city and season. For the Motor dataset, BS can enhance the performance on the edited attribute while maintaining the performance on the preserved attributes. Thirdly, we investigate the contribution of MR (Multi-Resolution). For the Synthetic and Air datasets, MR could improve the scores on all metrics. For the Motor dataset, MR helps improve the scores for the edited attributes and mostly sustains the performance on the preserved attributes. These improvements highlight the effectiveness of the proposed MR.

---

[3]Ground truth is only available in the Synthetic dataset.

## 4.4 Extended Investigation

We further conduct an in-depth analysis of the effectiveness and efficiency of our TEdit, aiming to reveal how it works through quantitative comparison and visualization.

**Editing vs. Conditional Generation.** Although conditional generation $\hat{\mathbf{x}}^{\text{tgt}} = \Phi_\theta(\emptyset, \emptyset, \mathbf{a}^{\text{tgt}})$ can produce time series that satisfy the specified target attribute values, it often struggles to retain sufficient details, particularly for difficult or unobservable attributes, since it generates samples from scratch. In contrast, editing $\hat{\mathbf{x}}^{\text{tgt}} = \Phi_\theta(\mathbf{x}^{\text{src}}, \mathbf{a}^{\text{src}}, \mathbf{a}^{\text{tgt}})$ is designed to modify the attributes of existing time series, and thus is able to preserve much detail information. We first quantitatively compare the two modes on the Synthetic data. The averaged MSE and MAE over all the finetuning datasets are presented in Tab. 3, which shows that editing could significantly outperform conditional generation with much better (lower) MSE and MAE scores.

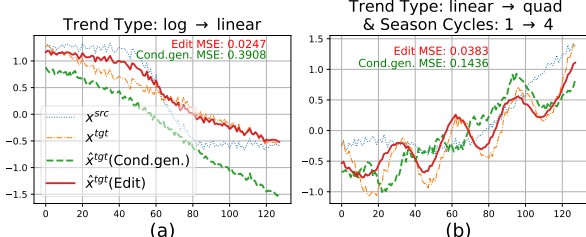

| Method | Mode | Synthetic | |
| | | MSE↓ | MAE↓ |
|---|---|---|---|
| TEdit-CSDI | Cond. Gen. | 0.2581 | 0.4096 |
| | Edit | **0.1235** | **0.2606** |
| TEdit-TW | Cond. Gen. | 0.2875 | 0.4308 |
| | Edit | **0.1315** | **0.2722** |

Figure 4: Case study for editing and conditional generation, under two settings: (a) editing the trend type from logistic to linear; (b) editing the trend type from linear to quadratic and the season cycles from 1 to 4.

Table 3: Quantitative comparison of editing and conditional generation on the Synthetic dataset. The results are averaged over all the finetuning sets.

We further showcase how editing and conditional generation work at the sample level. Fig. 4, plots the generated results of these two modes under. We can observe that the edited time series could better align with the target attributes, Fig. 4(a)(b), and preserve the characteristics of the source time series, e.g., bias in Fig. 4 (a). In comparison, conditional generation tends to produce smooth samples around the dataset mean. Additionally, we can also see in Fig. 4 that the MSE scores of editing are much lower than conditional generation. More visualization results can be found in Appendix I.4

**Efficiency of Multi-Resolution Generation.**
Considering that multi-resolution modeling and generation brings additional efforts for processing different resolutions, as discussed in Sec. 3.3. We compare the running time of serial processing and parallel processing implementations of the multi-resolution paradigm with total resolution number $R = 3$ and patch length $L_p^r = 3^r, r \in \{0, ..., R-1\}$. The experiment is conducted on the Synthetic dataset with a batch size of 32. The serial processing iterative processes a single resolution at a time, whereas the parallel processing concatenates multiple resolution sequences along the length dimension and processes them simultaneously. We record the time it takes for the full noise estimator and the processing module to complete the forward diffusion process. As shown in Tab. 4, our designed parallel mechanism can significantly improve the inference efficiency.

| Running time (ms) | Serial processing | Parallel processing |
|---|---|---|
| Noise estimator | 11.1 | 6.0 |
| Processing module | 8.8 | 3.7 |

Table 4: Running time (ms) of the multi-resolution modeling and generation module, averaged over 1000 samples.

**Bootstrap Improves the distributional coverage of the attribution space.** Fig. 5 visualizes the dimension reduced distribution of (a) the raw data, (b) the generated data and (c) the mixed data, on the Synthetic dataset using t-SNE [41]. We can observe that the newly generated data fulfill the uncovered region of the original raw data therefore enhance the data coverage over the whole space, which explains how the bootstrap learning helps generative model training. More visualization analysis can be found in Appendix I.5.

## 4.5 Sensitivity Analysis

In this subsection, we conduct sensitivity analysis to study the impacts of hyper-parameters of the multi-resolution mechanism and bootstrap learning.

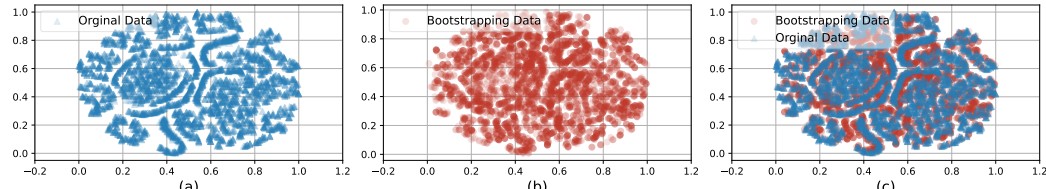

Figure 5: Visualization of data distribution before and after bootstrapping.

**Multi-resolution hyper-parameters.** We perform a sensitivity study for the hyper-parameters of the proposed multi-resolution, i.e., the patch length $L_p$ and the number of resolutions $R$, on one Synthetic finetuning set, which aims to edit trend directions while preserving other attributes. The results are presented in Fig. 6. More results on other Synthetic finetuning sets can be found in Appendix I.2.

To analyze the impact of $R$, we fix $L_p = 2$ and vary $R$. As shown in Fig. 6 (a), trend types prefer $R = 3$, whereas seasonal cycles perform better with $R = 4$. To examine the impact of $L_p$, we fix $R = 3$ and vary $L_p$. According to Fig. 6 (b), trend types show a preference for $L_p = 2$ while seasonal cycles prefer $L_p = 3$. Besides, we find that our proposed multi-resolution paradigm always performs better than the vanilla single resolution time-series modeling paradigms, i.e., $L_p = 1$ or $R = 1$, which corroborates the discussion in Sec. 3.3 that attributes may influence the time-series generation at different scales and granularities.

**Bootstrap learning hyper-parameters.** We conduct a sensitivity study on the hyper-parameters of bootstrapping, specifically the bootstrap ratio $\psi$ within the range $\{0.1, 0.3, 0.5, 0.7, 0.9\}$ on a Synthetic finetuning set, which aims to edit the trend direction while preserving others. As shown in Fig. 7, the model exhibits a poor performance then the bootstrap ratio is low, e.g., 0.1. The model has a good performance with medium to high bootstrap ratios. We hypothesize that this is because tuning with only a few top samples might lead to certain mode collapse.

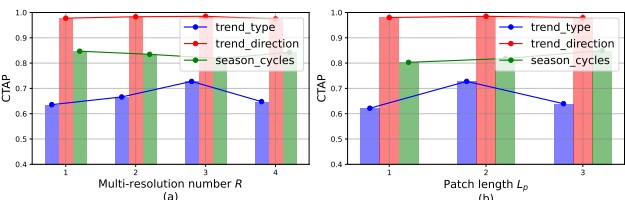 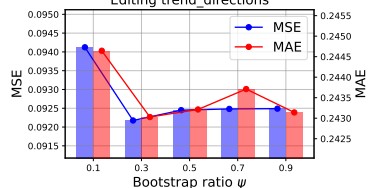

Figure 6: Sensitivity study for the hyper-parameters of multi-resolution on the Synthetic finetuning set for editing trend directions and preserving others. The higher CTAP, the better.

Figure 7: Sensitivity study of bootstrap ratio $\psi$ on a Synthetic finetuning set aiming to edit trend direction.

## 5 Conclusion

In this paper, we introduced the task of Time Series Editing (TSE), which enables controllable time series synthesis by manipulating specific attributes of the input time series while maintaining consistency in other properties. There are two major challenges of TSE. Firstly, in the real world, the distribution of time series and attributes is usually biased, and the full space might not be adequately covered. Secondly, time series and attributes exhibit complex multi-scale entanglement. To address these two challenges, we propose a novel diffusion based approach, called TEdit, which incorporates a bootstrap learning algorithm and a multi-resolution modeling and generation paradigm. Comprehensive experiments on both synthetic and real-world datasets demonstrate the effectiveness of our proposed TEdit in generating precise time series with specified attributes. In practice, our method still has certain limitations. For example, different attributes have varying difficulty degrees to edit; attributes may have complex inter-dependencies, and some attributes may be difficult to edit without affecting others. Despite this limitation, we believe our work lays the groundwork for controllable time series synthesis, potentially benefiting applications in fields such as climate monitoring, healthcare, and urban management.

## 6 Acknowledgment

This work is supported by HPC Platform of ShanghaiTech University, Shanghai Frontiers Science Center of Human-centered Artificial Intelligence, and MoE Key Lab of Intelligent Perception and Human-Machine Collaboration. We would like to express our sincere gratitude for their valuable support and resources that contributed significantly to the success of this research.

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

# A Related Work

**Time Series Generation.** Time series generation has become a research hotspot in recent years [47]. From the view of modeling, early attempts for time series generation utilize Generative Adversarial Networks (GANs) [47, 31], which trains the generator adversarial to the discriminator of correct classification on real or synthetic data. The latest research moves the focus to Diffusion models (DMs) [27, 36, 39] which have illustrated much higher fidelity and generation performance than GANs [45]. From the perspective of the generation paradigm, there are two branches. The first one is unconditional generation [47, 31, 34, 48], which employs a generative model to generate data samples without relying on any conditions for regulation. The second paradigm is conditional generation [39, 27, 36, 44] making use of additional conditional information like category labels, textual descriptions, features, and other kinds of metadata to guide the generation process and enabling high-quality generation that is tightly correlated with the conditions. Among them, CSDI [39] enhances time-series imputation by treating observed time series as conditions in the diffusion model. TIME WEAVER [27] directly encode the metadata into the diffusion model to achieve controlled generation. Though they achieved better quantitative results, these methods suffer from unexpected outcomes that inadvertently emphasize trivial attributes while overlooking critical ones, especially with a low availability of time-series data over various attributes.

**Diffusion Model.** Diffusion model [10, 37] has become increasingly prominent for data generation across various fields [26, 28, 2, 21], which learns to gradually regress and remove the noise that was added to the clean data in the forward process, allowing data generation aligned with real distribution. Our work is highly related to conditional generation, which can be divided into two main streams of techniques. Classifier-guided generation [6, 19, 24] adds the control signal from the gradient of the classifier to bias the generation towards the specific class in the denoising process of the diffusion model, which requires training an additional classification model. Another stream of works [11, 30] stem from classifier-free guidance in data generation, without training any additional classifier.

**Data Editing.** Data editing aims to manipulate the content of the given input to meet the requirements of the target data attributes, such as image style [17], semantics [25], and structure [7], with the utilization of multimodal methods such as text-to-image synthesis [9], [28, 49] and audio based image editing [50, 51]. In text-based scenarios, some works also generate texts under the instruction of specific attributes like sentiment [5] and structural information [32]. However, directly adopting the existing solution from other fields to time series editing is not feasible because of the significantly different data formats and poor data coverage of real-world time-series data w.r.t. different attributes.

# B Broader Impacts

Our method offers several potential positive societal impacts in many real-world scenarios. For example, in climate monitoring, it enhances climate models and forecasts, aiding in climate change mitigation. For urban management, it supports better decision-making in areas such as traffic management, pollution control, and resource allocation. Additionally, in research, it facilitates robust experiments and innovative solutions by providing high-quality, customizable datasets.

However, there are potential negative societal impacts to consider. There are data privacy risks associated with the manipulation of sensitive information, necessitating strong data protection measures. In financial markets, our method could be exploited for fraudulent activities, undermining market integrity. An overreliance on synthetic data may lead to biased or inaccurate models, resulting in flawed decision-making. Furthermore, there are ethical concerns regarding the potential misuse of our method in creating deceptive artificial intelligence models, which raises issues of transparency and accountability. While our method offers significant benefits, it is crucial to carefully manage these risks to maximize positive impacts and minimize negative consequences.

# C Diffusion Process

In this section, we give a review of the diffusion process. Diffusion models are a class of two-stage generation models, consisting of a forward process and a reverse(denoising) process. The forward process is a Markov chain which gradually adds random noise on the origin data sample $\mathbf{x}_0$ and

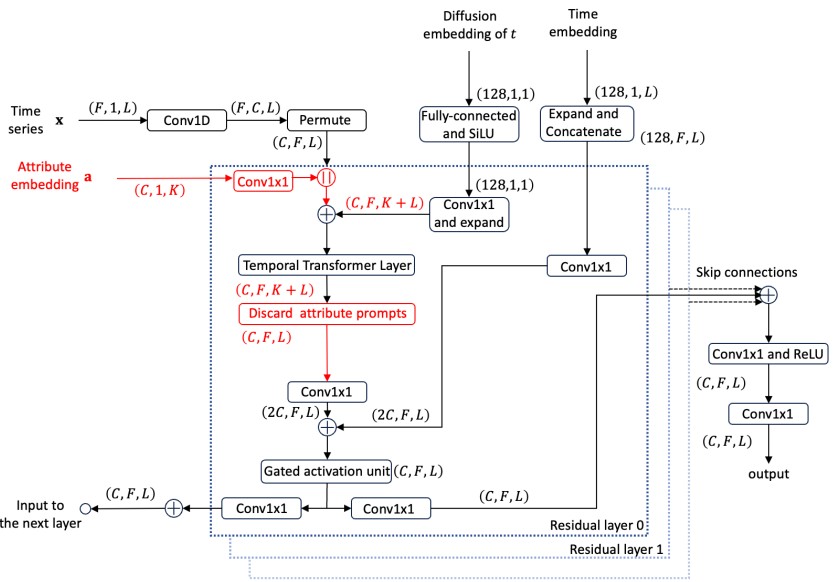

Figure 8: Model architecture of our modified CSDI.

intermediate latent variables $\mathbf{x}_t$ through a Gaussian transition $q(\mathbf{x}_t|\mathbf{x}_{t-1}) := \mathcal{N}(\sqrt{1-\beta_t}\mathbf{x}_{t-1}, \beta_t\mathbf{I})$, where $t = 1, \cdots, T$ indicates the diffusion step, and $\{\beta_t\}_{t=0}^{T}$ are predetermined variance schedule. The latent variable $\mathbf{x}_t$ can be described as:

$$\mathbf{x}_t = \sqrt{\alpha_t}\mathbf{x}_0 + \sqrt{1-\alpha_t}\boldsymbol{\epsilon}, \ \boldsymbol{\epsilon} \sim \mathcal{N}(0, \mathbf{I}), \ \alpha_t := \Pi_{s=1}^{t}(1-\beta_s). \tag{7}$$

In the reverse process, a denoiser $\boldsymbol{\epsilon}_\theta(\mathbf{x}_t, t, \mathbf{a})$, where $\mathbf{a} \in \mathbb{N}^K$ is the given attributes, is needed to approximate the noise in each diffusion step to help to sample backward from $\mathbf{x}_T$ through another Gaussian transition $p_\theta(\mathbf{x}_{t-1}|\mathbf{x}_t) := \mathcal{N}(\mathbf{x}_{t-1}; \mu_\theta(\mathbf{x}_t, t), \sigma_\theta(\mathbf{x}_t, t)\mathbf{I})$. To obtain $\mu_\theta(\mathbf{x}_t, t)$, the denoiser should be trained by optimizing the loss:

$$\min_\theta \mathcal{L} := \min_\theta \mathbb{E}_{\mathbf{x}_0 \sim q(\mathbf{x}_0), \boldsymbol{\epsilon} \sim \mathcal{N}(\mathbf{0}, \mathbf{I}), t \sim \mathcal{U}(1, T)} \|\boldsymbol{\epsilon} - \boldsymbol{\epsilon}_\theta(\mathbf{x}_t, t, \mathbf{a})\|_2^2 \tag{8}$$

Here $q(\mathbf{x}_0)$ denotes the data distribution of $\mathbf{x}_0$ (or $\mathbf{x}$), $t \sim \mathcal{U}(1, T)$ indicates that the diffusion step $t$ follows the uniform distribution between $1$ and $T$.

After training, given a condition $\mathbf{a}$, we can generate the sample $\hat{\mathbf{x}}_0$ from a random noise $\hat{\mathbf{x}}_T \sim \mathcal{N}(0, \mathbf{I})$ using the trained $\boldsymbol{\epsilon}_\theta(\mathbf{x}_t, t, \mathbf{a})$ with a diffusion sampler, such as the deterministic denoising diffusion implicit model (DDIM) [38]:

$$\hat{\mathbf{x}}_{t-1} = \sqrt{\alpha_{t-1}}\mathbf{f}_\theta(\hat{\mathbf{x}}_t, t, \mathbf{a}) + \sqrt{1-\alpha_{t-1}}\boldsymbol{\epsilon}_\theta(\hat{\mathbf{x}}_t, t, \mathbf{a}) \tag{9}$$

$$\mathbf{f}_\theta(\hat{\mathbf{x}}_t, t, \mathbf{a}) = (\hat{\mathbf{x}}_t - \sqrt{1-\alpha_t}\boldsymbol{\epsilon}_\theta(\hat{\mathbf{x}}_t, t, \mathbf{a}))/\sqrt{\alpha_t} \tag{10}$$

where $\mathbf{f}_\theta(\hat{\mathbf{x}}_t, t, \mathbf{a})$ can be regarded as a prediction of $\mathbf{x}_0$ at the diffusion step $t$.

## D Model Architecture

In this section, we will show the architecture utilized in our modified backbone models CSDI 8 and TIME-WEAVER 9, especially how the attributes are incorporated into the models. It should be mentioned that all the attributes in our experiment setting are categorical.

Consider a batch of time series samples of size $(N_{\text{batch}}, F, L)$, where $N_{\text{batch}}$ represents the number of samples per batch, $F$ represents the number of channels in the time series, and $L$ represents the horizon. The corresponding attributes $\mathbf{a}$ is of shape $(N_{\text{batch}}, L, K)$, where $K$ represents the number

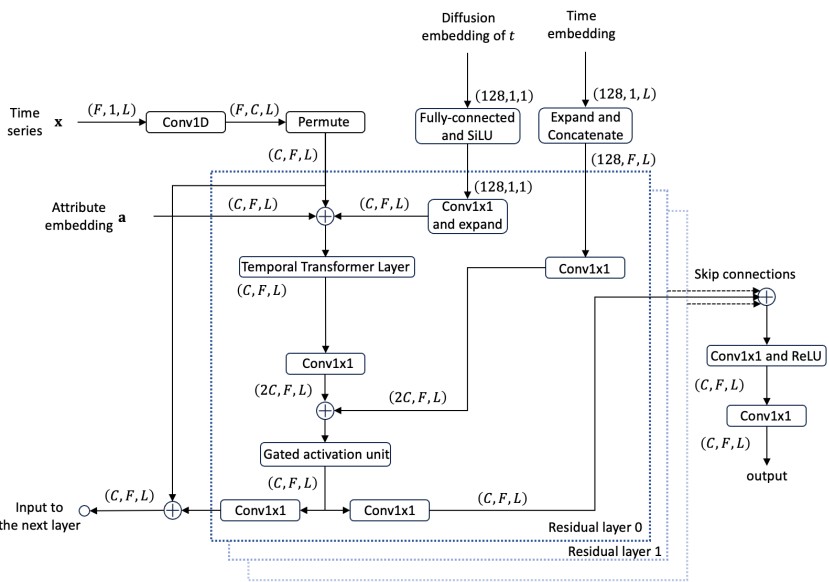

Figure 9: Model architecture of Time Weaver.

of the attributes. While performing the time-series inputs, each attribute option will be encoded to a corresponding embedding vector.

As for CSDI, our modified part is highlighted in red. First, the attribute embedding will be linearly projected to shape $(N_{\text{batch}}, d, F, K)$, then concatenated with the time series embedding to form the prompt with shape $(N_{\text{batch}}, d, F, K + L)$. Next, the prompt is added with the expanded diffusion step embedding and the sum will be inputted into the transformer layer. After that, the attribute embedding will be discarded, and the shape gets back to $(N_{\text{batch}}, d, F, L)$.

For TIME-WEAVER, We only make some slight changes to its original structure: 1. Our experiment setting doesn't include continuous type attributes, as mentioned. 2. We replace the self-attention modules with multi-layer perceptrons (MLPs).

As mentioned in Sec 3.3, we introduce a novel parallelization with an attention masking mechanism to accelerate the training and inference. The attention mask is of the following form, denoted as $\mathbf{M} \in \mathbb{R}^{(N_p^0 + \dots + N_p^{R-1}) \times (N_p^0 + \dots + N_p^{R-1})}$ where $\mathbf{M}^r \in \mathbf{0}^{N_p^r}$, the residual value is negative infinity. Through adding this attention mask into the original attention calculation [42] and softmax operation $\text{softmax}(\mathbf{z})_i = \frac{e^{z_i}}{\sum_{j=1}^{N} e^{z_j}}$ receiving $\mathbf{z} \in \mathbb{R}^N$, such attention mask would mask out inter-sequence attention while preserving intra-sequence attention.

$$\mathbf{M} = \begin{pmatrix} \mathbf{M^1} & \cdots & -\infty \\ \vdots & \ddots & \vdots \\ -\infty & \cdots & \mathbf{M^{R-1}} \end{pmatrix}$$

## E  Bootstrap Learning Algorithm

As mentioned in Sec. 3.4, we propose a bootstrap learning algorithm that enables the model $\epsilon_\theta$ to learn the information of the imaginary target. Given the pretained model $\epsilon_\theta$, the detailed algorithm for a batch $(\mathbf{X}_0^{\text{src}} \in \mathbb{R}^{B \times L}, \mathbf{A}^{\text{src}} \in \mathbb{N}_+^{B \times K})$, where $B$ is the batch size, $L$ is the length, and $K$ is the number of attributes, is presented below.

**Algorithm 1** Bootstrap Learning for a Batch

---

**Input:** $(\mathbf{X}_0^{\text{src}} \in \mathbb{R}^{B \times L}, \mathbf{A}^{\text{src}} \in \mathbb{N}_+^{B \times K})$, where $B$, $L$, $K$ are batch size, length, and attribute number.
Attributes to be edited $\mathcal{A}_{\text{edit}}$ and preserved $\mathcal{A}_{\text{prsv}}$.
The ratio $\psi$ of the top bootstrapped samples to include for tuning.

1: **# 1. Compose tuple $(\mathbf{X}_0^{\text{src}}, \mathbf{A}^{\text{src}}, \mathbf{A}^{\text{tgt}})$.**
2: **for** $k = 1, \cdots, K$ **do**
3:    **if** $k \in \mathcal{A}_{\text{edit}}$ **then**
4:       $\mathbf{A}^{\text{tgt}}[:, k] \sim \mathcal{U}\{1, \cdots, N_k\}$          ▷ *Random sample a new value for the k-th attribute.*
5:    **else**
6:       $\mathbf{A}^{\text{tgt}}[:, k] = \mathbf{A}^{\text{src}}[:, k]$

7: **# 2. Generate the target $\hat{\mathbf{X}}^{\text{tgt}}$.**
8: $\hat{\mathbf{X}}_0^{\text{tgt}} \leftarrow \Phi_\theta(\mathbf{X}_0^{\text{src}}, \mathbf{A}^{\text{src}}, \mathbf{A}^{\text{tgt}})$

9: **# 3. Self-Score the target $\hat{\mathbf{X}}_0^{\text{tgt}}$, and keep the top $\psi$ samples.**
10: $\hat{\mathbf{X}}_0^{\text{src}} \leftarrow \Phi_\theta(\hat{\mathbf{X}}_0^{\text{tgt}}, \mathbf{A}^{\text{tgt}}, \mathbf{A}^{\text{src}})$
11: $\mathbf{s} \leftarrow \text{MSE}(\mathbf{X}_0^{\text{src}}, \hat{\mathbf{X}}_0^{\text{src}})$
12: $\hat{\mathbf{X}}_0^{\text{tgt}} \leftarrow \text{Sort}(\hat{\mathbf{X}}_0^{\text{tgt}} | \mathbf{s})$          ▷ *Sort $\hat{\mathbf{X}}_0^{tgt}$ based on $\mathbf{s}$.*
13: $\hat{\mathbf{X}}_{0,\text{bs}}^{\text{tgt}} \leftarrow \hat{\mathbf{X}}_0^{\text{tgt}}[: B \cdot \psi]$          ▷ *Keep the top $\psi$ samples.*

14: **# 4. Calculate the loss.**
15: $\mathcal{L}_{BS} \leftarrow \mathcal{L}(\hat{\mathbf{X}}_{0,\text{bs}}^{\text{tgt}})$      ▷ *$\mathcal{L}$ is the noise estimation loss of diffusion models defined in Eq.(1).*

| Attribute | Value |
|---|---|
| Trend Types | [Linear, Quadratic, Exponential, Logistic] |
| Trend Directions | [Up, Down] |
| Season Cycles | [0, 1, 2, 4] |

Table 5: Summary of attribute options

# F  Datasets

## F.1  Synthetic Dataset

We generate the synthetic time series according to the following formula:

$$\mathbf{x} = \mathbf{x}_{\text{trend}} + \mathbf{x}_{\text{season}} + \mathbf{x}_{\text{noise}} + \mathbf{x}_{\text{bias}}. \tag{11}$$

Table 5 shows the summary of the attributes for the synthetic dataset.

For example, the attribute {Trend Type = linear, Trend Direction = up, Season Cycles = 2} is one of the total 32 (4 Trend Types × 2 Trend Directions × 4 Season Cycles) attribute combinations. We split the 32 combinations into train, validation, and test sets by 24:4:4.

In the pretraining dataset, for each attribute combination, we sample 300 time series according to the random variances described in the following subsections. For the finetuning dataset, we treat each attribute combination in the 24:4:4 split as the target attribute. For each target attribute, we compose its source attribute based on the desired attributes to be edited $\mathcal{A}_{\text{edit}}$ and preserved $\mathcal{A}_{\text{prsv}}$. For example, let $\mathcal{A}_{\text{edit}} = \{\text{trend type}\}$, $\mathcal{A}_{\text{prsv}} = \{\text{trend direction}, \text{season cycles}\}$. Suppose the target attribute $\mathbf{a}^{\text{tgt}} = \{\text{linear}, \text{up}, 2\}$, then its source could be $\mathbf{a}_1^{\text{src}} = \{\text{linear}, \text{up}, 2\}$, $\mathbf{a}_2^{\text{src}} = \{\text{quadratic}, \text{up}, 2\}$, $\mathbf{a}_3^{\text{src}} = \{\text{exponential}, \text{up}, 2\}$ and, $\mathbf{a}_4^{\text{src}} = \{\text{logistic}, \text{up}, 2\}$. Then for a source and target attribute pair $(\mathbf{a}^{\text{src}}, \mathbf{a}^{\text{tgt}})$, we generate the source and target time series $(\mathbf{x}^{\text{src}}, \mathbf{x}^{\text{tgt}})$ based on the process described in the following subsections. Note that for each $(\mathbf{a}^{\text{src}}, \mathbf{a}^{\text{tgt}})$ attribute pair, we first generate the deterministic part, e.g., trend type, for $\mathbf{a}^{\text{src}}$ and $\mathbf{a}^{\text{tgt}}$, denoted by $\mathbf{x}_{deter}^{\text{src}}$ and $\mathbf{x}_{deter}^{\text{tgt}}$. Then generate 10 randomnesses $\mathbf{x}_{rand}$, including noise, bias etc, and then we add these randomnesses to the

deterministic part to ensure the source and target time series have the same randomness $\mathbf{x}_{rand}$: $\mathbf{x}^{\text{tgt}} = \mathbf{x}^{\text{tgt}}_{deter} + \mathbf{x}_{rand}$, $\mathbf{x}^{\text{src}} = \mathbf{x}^{\text{src}}_{deter} + \mathbf{x}_{rand}$.

The finetuning dataset contains 6 subsets for editing different attribute combinations. For modifying trend type, there are 760 train samples, 140 validation samples, and 160 test samples. For modifying trend direction, there are 420 train samples, 70 validation samples, and 80 test samples. For modifying season cycles, there are 760 train samples, 140 validation samples, and 160 test samples. For modifying trend type and trend direction, there are 1440 train samples, 280 validation samples, and 320 test samples. For modifying trend type and season cycles, there are 2880 train samples, 560 validation samples, and 640 test samples. For modifying trend direction and season cycles, there are 1440 train samples, 280 validation samples, and 320 test samples. In addition, the same random variance and bias pair is used both on the source time series and target time series in a source-target pair respectively.

In the following formulas, $t_i$ indicates the $i$-th sampling point in the original synthetic time series($\mathbf{t}$) and $x_i$ indicates the $i$-th sampling point in a contain form of $\mathbf{x}$. For instance, the total length of a time series $L = 128$, then $i = [1, \ldots, 128]$.

### F.1.1   Trend

**Trend Types.** As described in table 5, there are 4 trend types: linear, quadratic, exponential, and logistic. As described above, $\mathbf{t}$ is used to obtain $\mathbf{x}$. For the **linear** trend: $\mathbf{x}_{\text{trend}} = \mathbf{t}$, in this case $t_i \in [0, 1]$, $x_i \in [0, 1]$. For the **quadratic** trend: $\mathbf{x}_{\text{trend}} = \mathbf{t}^2$, in this case $t_i \in [0, 1]$, $x_i \in [0, 1]$. For the **exponential** trend: $\mathbf{x}_{\text{trend}} = \frac{2^{\mathbf{t}}}{1024}$, in this case $t_i \in [-10, 10]$, $x_i \in [0, 1]$. $\mathbf{x}_{\text{trend}}$ is needed to range from 0 to 1, so $t_i$ is in [-10,10]. For **logistic** trend: $\mathbf{x}_{\text{trend}} = \frac{1}{1+\exp(-\mathbf{t})}$, where $t_i \in [-10, 10]$, $x_i \in [0, 1]$. Similar to the exponential trend, we repeat a scaling process in logistic trend. To train the model more easily, $\mathbf{x}_{\text{trend}} = (\mathbf{x} - 0.5) \times 2$ is used to normalize $\mathbf{x}_{\text{trend}}$ to $[-1, 1]$.

**Trend Directions.** There are totally 2 directions: up and down. For instance, in the Cartesian coordinate system, a linear line from coordinates $(0, 0)$ to $(1, 1)$ represents an "up" trend, while another line from coordinates $(0, 0)$ to $(1, -1)$ represents a "down" trend. All 4 trend types introduce above are originally "up" trends. Therefore, for an "up" trend, we set $\mathbf{x}_{\text{trend}} = \mathbf{x}_{\text{trend}}$ and for a "down" trend, we set $\mathbf{x}_{\text{trend}} = -\mathbf{x}_{\text{trend}}$.

**Random Variance.** We randomly perturb the scale of the trend by: $\mathbf{x}_{\text{trend}} = s \cdot \mathbf{x}_{\text{trend}}$ where $s$ is the random scale obtained by the following process $s = m \cdot c + (1 - m) \cdot \frac{1}{c}$, $m \sim Bern(0.5)$, $c \sim \mathcal{U}(0.8, 1.0)$ If $m = 1$, then $s = c \in [0.8, 1.0]$; else $s = \frac{1}{c} \in [1.0, 1.25]$.

### F.1.2   Season

The season attribute is represented by the number of cycles in a time series. [0,1,2,4] sinusoidal wave cycles are randomly added.

$$\mathbf{x}_{\text{season}} = a \sin(2\pi t + \phi) \tag{12}$$

where $t \in [0, n_{\text{cycle}}], n_{\text{cycle}} \in [0, 2^0, 2^1, 2^2]$. The Random Variances of these two variables follow uniform distributions: $a \sim \mathcal{U}(0.4, 0.6)$, $\phi \sim \mathcal{U}(0, 2\pi)$.

### F.1.3   Noise

We simulate the real-world noise by a combination of Gaussian noise and high-frequency noise.

$$\mathbf{x}_{\text{noise}} = \mathbf{x}_{\text{g}} + \mathbf{x}_{\text{hf}} \tag{13}$$

The Gaussian noise is sampled through $\mathbf{x}_{\text{g}} = \mathcal{N}(0, \sigma), \sigma \sim \mathcal{U}(0.04, 0.06)$. The high-frequency noise is portrayed by a sinusoidal wave: $\mathbf{x}_{\text{hf}} = a \sin(2\pi t + \phi)$, $t \in [0, n_{\text{cycle}}]$, $n_{\text{cycle}} \in [2^4, 2^5, 2^6]$, $a \sim \mathcal{U}(0.08, 0.10)$, $\phi \sim \mathcal{U}(0, 2\pi)$.

### F.1.4   Bias

The last step is to add bias to the previous synthesized sample, which is randomly sampled from a uniform distribution $\mathbf{x}_{\text{bias}} \sim \mathcal{U}(-0.5, 0.5)$.

| Dataset | L | Categorical and Editable Attributes |
|---------|---|-------------------------------------|
| Air Quality [8] | 168 | 4 Seasons [Spring, Summer, Autumn, Winter] 2 Cities [Beijing, London] |
| Motor Imagery [1] | 150 | 2 Movements [Figure, Tongue] 64 Channels[1, 2, ..., 64] |

Table 6: Dataset description in our experiments. Some key information is listed above. The real-world datasets include Air Quality, Motor Imagery. All datasets have been carefully processed. $L$ indicates the time series length. The second column indicates all editable attributes.

## F.2 Real World Datasets

### F.2.1 Air Quality Dataset

This dataset was used in the KDD Cup 2018 forecasting competition [8]. It contains lots of hourly sampled time series representing the air quality levels measured from different stations in 2 cities: Beijing (35 stations) and London (24 stations) from 01/01/2017 to 31/03/2018. This air quality level is represented in multiple measurements such as PM2.5, PM10, $NO_2$, CO, $O_3$, and $SO_2$.

We processed the raw data and split it into train, validation, and test sets. We only selected the data of PM2.5, which is common in all stations. Season attributes are generated according to month. We slice the series by week to a length of 168 (24hour $\times$ 7day). For pretraining dataset, there are totally 3650 time series samples, in which we randomly pick 2825 time series samples as train, 353 time series samples as validation, and 472 time series samples as the test. For finetuning dataset, it's composed of multi-subsets where each subset corresponds to an editing setting, e.g. modifying the "season". In any subset, a sample is composed of a source time series, source attributes, and target attributes where the source time series are sampled from the corresponding splits of pretraining data. For each source time series, we randomly change the source attributes 2 or 3 times to get the target attributes and pair the source time series and target attributes together. Finally, we get (train: 2000, valid: 600, test: 600) samples for modifying "city" subset and (train: 3000, validation: 900, test: 900) samples for modifying "season" subset.

### F.2.2 Motor Imagery Dataset

The Motor Imagery dataset [1] contains some Electroencephalogram (EEG) records. During the Brain-Computer Interface experiments, the subjects have to perform imagined movements of either the tongue or the left small finger. All recordings are collected at a sampling rate of 1000 Hz with 64 channels. Every record contains 3000 time stamps (3 seconds measurement). For pertaining data, there are 19353 samples in the train set, 2419 in the validation set, 2419 in the test set. For fine-tuning data, the processing operation is similar to the Air Quality. In the subset of modifying movement, there are (train:2000, valid:1000, test:1000) samples. In the subset of modifying channel id, there are (train:3000, valid:1500, test:1500) samples.

## G CTAP Model Details

Our Contrastive Time-Series Attribute Pretraining (CTAP) model is similar to the popular CLIP model [33]. The purpose of this model is to train the time series encoder and the attribute encoder by learning the alignment between the time series $\mathbf{x} \in \mathbb{R}^L$ and its associated attributes $\mathbf{a} \in \mathbb{N}_+^K$. Note that we use separate attribute encoders for different attributes $a_k = \mathbf{a}[k]$, $k \in \{1, ..., K\}$. An illustration of CTAP for a batch of $\mathbf{X} \in \mathbb{R}^{B \times L}$ and their associated $k$-th attribute $\mathbf{A} \in \mathbb{N}_+^B$ is presented in Fig. 10. Note that the attribute index $k$ is dropped for clarity. The time series encoder and attribute encoder extract the embeddings $\mathbf{H_x} = \{\mathbf{h}_{\mathbf{x}_i}\}_{i=1}^B$ and $\mathbf{H}_a = \{\mathbf{h}_{a_i}\}_{i=1}^B$. Following [33], we calculate the pair-wise similarities between the $\mathbf{h}_{\mathbf{x}_i}$ and $\mathbf{h}_{a_j}$, and the encoders are trained by distinguishing whether $\mathbf{h}_{\mathbf{x}_i}$ and $\mathbf{h}_{a_j}$ are from the same data pair. The pseudocode of the CTAP model is shown in Algorithm 2.

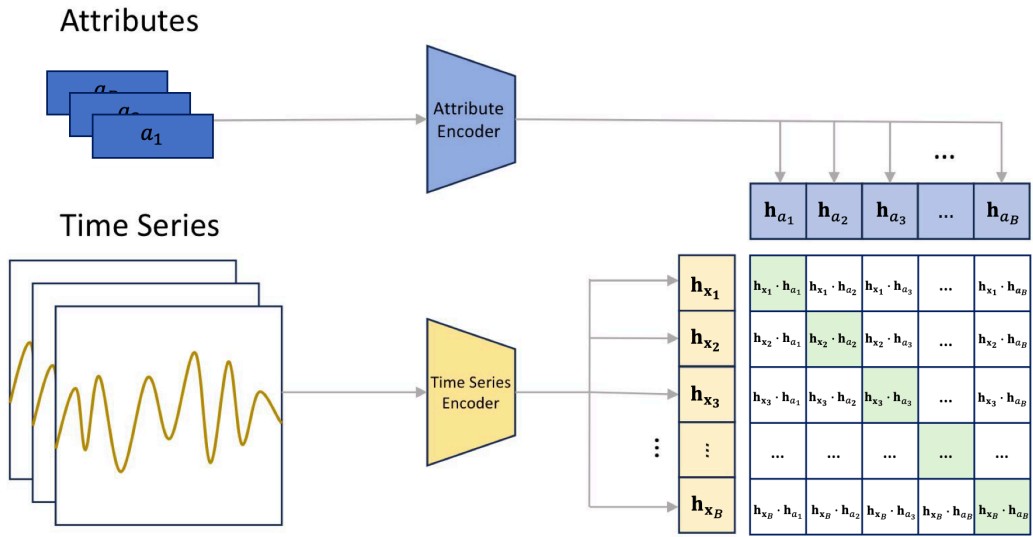

Figure 10: Illustration of the CTAP model for the given pairs of time series $\mathbf{X} = \{\mathbf{x}_i\}_{i=1}^B$ and attributes $\mathbf{A} = \{a_i\}_{i=1}^B$, where $B$ is the batch size. Here $a = a_k = \mathbf{a}[k]$, $k \in \{1, ..., K\}$ is the $k$-th attribute of the full attribute vector $\mathbf{a} \in \mathbb{N}_+^K$. We use $K$ separate attribute encoders for $K$ attributes. In the illustration, we only show one attribute and thus drop the attribute index $k$ for clarity. After obtaining embeddings $\{\mathbf{h}_{\mathbf{x}_i}\}_{i=1}^B$ and $\{\mathbf{h}_{a_i}\}_{i=1}^B$, we calculate the the pair-wise similarities between the $\mathbf{h}_{\mathbf{x}_i}$ and $\mathbf{h}_{a_j}$. The encoders are trained by distinguishing the positive pairs (green blocks) and negative pairs (white blocks).

---

**Algorithm 2** Pseudocode for the CTAP model

---

**Input:** A batch of paired time series and attributes ($\mathbf{X} \in \mathbb{R}^{B \times L}$, $\mathbf{A} \in \mathbb{R}^B$).

1: **# 1. Extract embeddings of time series and attributes.**
2: $\mathbf{H_x} \leftarrow$ Time Series Encoder($\mathbf{X}$)
3: $\mathbf{H}_a \leftarrow$ Attribute Encoder($\mathbf{A}$)

4: **# 2. Calculate pairwise similarities.**
5: $\mathbf{S} \leftarrow \text{Sim}(\mathbf{H_x}, \mathbf{H}_a)$          *▷ $\mathbf{S} \in \mathbb{R}^{B \times B}$ is the similarity score matrix.*

6: **# 3. Calculate loss.**
7: $\mathcal{L}_\mathbf{x} \leftarrow$ Cross Entropy($\mathbf{S}, \mathbf{I}$, axis = 1)          *▷ $\mathbf{I} \in \{0, 1\}^{B \times B}$ is the identity matrix.*
8: $\mathcal{L}_a \leftarrow$ Cross Entropy($\mathbf{S}, \mathbf{I}$, axis = 0)
9: $\mathcal{L} \leftarrow (\mathcal{L}_\mathbf{x} + \mathcal{L}_a)/2$

---

| | Synthetic | | | Air | | Motor | |
|---|---|---|---|---|---|---|---|
| | Trend types | Trend directions | Season cycles | City | Season | Channel id | Imagined movement |
| Top1 Acc | 0.7877 | 1.0000 | 0.9558 | 0.9725 | 0.9068 | 0.8979 | 0.9537 |
| Top2 Acc | 0.9408 | 1.0000 | 0.9983 | 1.0000 | 0.9492 | 0.9364 | 1.0000 |

Table 7: The performance of CTAP models on different datasets. We report the top-1 and top-2 classification accuracy for each attribute on the test sets of the pertaining dataset.

# H Evaluation Metrics

## H.1 MSE and MAE

For the synthetic dataset, MSE and MAE are two of the metrics used in our experiments. For the reason that we have targets in our synthetic datasets, MSE and MAE can be applied to evaluate the generated time series. The two formulas are listed below:

$$\text{MSE} = \frac{1}{|\mathcal{D}|} \sum_{i=1}^{|\mathcal{D}|} (\mathbf{x}_i - \hat{\mathbf{x}}_i)^2 \tag{14}$$

$$\text{MAE} = \frac{1}{|\mathcal{D}|} \sum_{i=1}^{|\mathcal{D}|} |\mathbf{x}_i - \hat{\mathbf{x}}_i| \tag{15}$$

Here $|\mathcal{D}|$ is the size of the test set, $\mathbf{x}_i$ is the $i$-th target time series, and $\hat{\mathbf{x}}_i$ is the corresponding generated time series.

## H.2 CTAP

The CTAP score is obtained through the Contrastive Time series - Attribute Pretraining (CTAP) model, which is similar to the popular CLIP model [33]. The details of the CTAP model are given in Appendix G. Suppose $a_k^{\text{tgt}}$ is one of the target attributes, the CTAP score is similar to the CLIP-I score [23], which uses the pretrained CTAP model to measure the alignment, e.g., cosine similarity, between the generated $\hat{\mathbf{x}}^{\text{tgt}}$ and the real-world time series that also associated with the attribute $a_k^{\text{tgt}}$. Specifically, we first use the CTAP model to extract embeddings $\hat{\mathbf{h}}_{\mathbf{x}}$, which is the embedding of $\hat{\mathbf{x}}^{\text{tgt}}$, and $\bar{\mathbf{h}}_{\mathbf{x}}|a_k^{\text{tgt}}$, which is the average embedding for the time series associated with $a_k^{\text{tgt}}$ in the training data. Then we calculate the cosine similarity of $\hat{\mathbf{h}}_{\mathbf{x}}$ and $\bar{\mathbf{h}}_{\mathbf{x}}|a_k^{\text{tgt}}$, where the higher similarity indicates better alignment between the generated and real-world time series.

## H.3 RaTS

We introduce Log *Ra*tio of *T*arget-to-*S*ource probability (RaTS) score to measure whether the generated time series $\hat{\mathbf{x}}^{\text{tgt}}$ is closer to the target attribute $a_k^{\text{tgt}}$ than the source time series $\mathbf{x}^{\text{src}}$. Formally, the RaTS score for a tuple $(\hat{\mathbf{x}}^{\text{tgt}}, \mathbf{x}^{\text{src}}, a_k^{\text{tgt}})$ is defined as:

$$\text{RaTS}(\hat{\mathbf{x}}^{\text{tgt}}, \mathbf{x}^{\text{src}}, a_k^{\text{tgt}}) = \log\left(\frac{p(a_k^{\text{tgt}}|\hat{\mathbf{x}}^{\text{tgt}})}{p(a_k^{\text{tgt}}|\mathbf{x}^{\text{src}})}\right). \tag{16}$$

where $p(a_k|\mathbf{x})$ is calculated by applying a softmax over the similarity scores of all $(\mathbf{h}_{\mathbf{x}}, \mathbf{h}_{a_k=i})$ pairs, where $\mathbf{h}_{\mathbf{x}}$ is the CTAP embedding of $\mathbf{x}$, and $\mathbf{h}_{a_k=i}$, $i \in \{1, ..., N_k\}$, is the CTAP embedding of the $i$-th possible value for the $k$-th attribute $a_k$.

RaTS > 0 represents the model doesn't retain characteristics of the source time series $\mathbf{x}^{\text{src}}$ to a certain degree. RaTS < 0 means that the attribute $a_k^{\text{tgt}}$ is reinforced compared to the source $\mathbf{x}^{\text{src}}$. For the edited attributes $\mathcal{A}_{\text{edit}}$, the higher the RaTS score, the closer the edited $\hat{\mathbf{x}}^{\text{tgt}}$ is to the target attributes. For the preserved attributes $\mathcal{A}_{\text{prsv}}$, the lower the |RaTS| score, the better the preservation of the attributes.

# I  More Experimental Results

## I.1  Results

|  | Overall | | Trend Type▲ | | Trend Direction△ | | Season Cycles△ | |
|---|---|---|---|---|---|---|---|---|
|  | ↓MSE | ↓MAE | ↑RaTS | ↑CTAP | ↓\|RaTS\| | ↑CTAP | ↓\|RaTS\| | ↑CTAP |
| CSDI | 0.1049±0.0852 | 0.2334±0.0988 | 0.5661±0.1651 | 0.3888±0.1777 | **0.0014**±0.0001 | **0.9877**±0.0004 | **0.0621**±0.0049 | 0.8337±0.0307 |
| Time Weaver | 0.0519±0.0257 | 0.1783±0.0422 | 0.8460±0.1394 | 0.6870±0.1197 | 0.0016±0.0002 | 0.9871±0.0004 | 0.1012±0.0219 | 0.8219±0.0261 |
| TEdit-CSDI | **0.0377**±0.0065 | **0.1478**±0.0121 | 0.8442±0.0810 | 0.6775±0.0701 | 0.0017±0.0004 | 0.9869±0.0013 | 0.0813±0.0135 | **0.8562**±0.0128 |
| TEdit-TW | 0.0522±0.0141 | 0.1817±0.0272 | **0.9006**±0.0707 | **0.7205**±0.0542 | 0.0015±0.0001 | 0.9874±0.0008 | 0.1042±0.0141 | 0.8382±0.0181 |

|  | Overall | | Trend Type△ | | Trend Direction▲ | | Season Cycles△ | |
|---|---|---|---|---|---|---|---|---|
|  | ↓MSE | ↓MAE | ↓\|RaTS\| | ↑CTAP | ↑RaTS | ↑CTAP | ↓\|RaTS\| | ↑CTAP |
| CSDI | 0.2128±0.1502 | 0.3726±0.1359 | 0.6227±0.2207 | 0.3655±0.2287 | 1.9692±0.0010 | 0.9748±0.0073 | **0.0542**±0.0158 | **0.8641**±0.0062 |
| Time Weaver | 0.1345±0.0262 | 0.2989±0.0320 | 0.3564±0.1016 | 0.6309±0.1071 | 1.9711±0.0009 | 0.9833±0.0030 | 0.0918±0.0289 | 0.7877±0.0329 |
| TEdit-CSDI | 0.0979±0.0185 | 0.2509±0.0286 | 0.3476±0.1117 | 0.6673±0.1024 | 1.9717±0.0015 | 0.9824±0.0019 | 0.0660±0.0090 | 0.8418±0.0263 |
| TEdit-TW | **0.0924**±0.0202 | **0.2432**±0.0269 | **0.2581**±0.0530 | **0.7277**±0.0293 | **1.9719**±0.0007 | **0.9849**±0.0028 | 0.0753±0.0148 | 0.8182±0.0123 |

|  | Overall | | Trend Type△ | | Trend Direction△ | | Season Cycles▲ | |
|---|---|---|---|---|---|---|---|---|
|  | ↓MSE | ↓MAE | ↓\|RaTS\| | ↑CTAP | ↓\|RaTS\| | ↑CTAP | ↑RaTS | ↑CTAP |
| CSDI | **0.1980**±0.0105 | **0.3508**±0.0141 | **0.1174**±0.0451 | 0.6962±0.0754 | **0.0012**±0.0001 | 0.9875±0.0008 | 0.3758±0.0840 | 0.0944±0.0862 |
| Time Weaver | 0.2167±0.0186 | 0.3632±0.0107 | 0.1767±0.0238 | 0.7221±0.0632 | 0.0019±0.0005 | 0.9847±0.0023 | 0.5976±0.0903 | 0.3233±0.0804 |
| TEdit-CSDI | 0.2054±0.0149 | 0.3538±0.0229 | 0.1390±0.0436 | 0.7795±0.0668 | 0.0013±0.0000 | 0.9877±0.0014 | 0.6564±0.2560 | 0.3670±0.2543 |
| TEdit-TW | 0.2122±0.0228 | 0.3623±0.0208 | 0.1652±0.0305 | **0.8021**±0.0150 | 0.0013±0.0001 | **0.9881**±0.0005 | **0.8911**±0.0688 | **0.6045**±0.0668 |

|  | Overall | | Trend Type▲ | | Trend Direction△ | | Season Cycles▲ | |
|---|---|---|---|---|---|---|---|---|
|  | ↓MSE | ↓MAE | ↑RaTS | ↑CTAP | ↑RaTS | ↑TAP | ↓\|RaTS\| | ↑CTAP |
| CSDI | 0.1119±0.0631 | 0.2532±0.0777 | 0.4925±0.1530 | 0.4653±0.1492 | 1.1210±0.0005 | 0.9835±0.0032 | **0.0520**±0.0071 | 0.8546±0.0081 |
| Time Weaver | 0.0712±0.0160 | 0.2109±0.0264 | 0.7017±0.0986 | 0.6846±0.0893 | 1.1216±0.0003 | **0.9872**±0.0003 | 0.0798±0.0183 | 0.8277±0.0252 |
| TEdit-CSDI | **0.0451**±0.0085 | **0.1638**±0.0138 | 0.7478±0.0754 | 0.7215±0.0693 | 1.1216±0.0004 | 0.9864±0.0014 | 0.0544±0.0028 | **0.8609**±0.0080 |
| TEdit-TW | 0.0676±0.0254 | 0.1997±0.0359 | **0.8014**±0.0629 | **0.7706**±0.0507 | **1.1222**±0.0002 | 0.9870±0.0002 | 0.0670±0.0092 | 0.8489±0.0087 |

|  | Overall | | Trend Type▲ | | Trend Direction△ | | Season Cycles▲ | |
|---|---|---|---|---|---|---|---|---|
|  | ↓MSE | ↓MAE | ↑RaTS | ↑CTAP | ↓\|RaTS\| | ↑CTAP | ↑RaTS | ↑CTAP |
| CSDI | 0.2065±0.0708 | 0.3446±0.0607 | 0.4990±0.1734 | 0.5078±0.1775 | 0.0022±0.0003 | 0.9840±0.0024 | 0.3325±0.0944 | 0.2721±0.0921 |
| Time Weaver | 0.1843±0.0341 | 0.3286±0.0323 | 0.6915±0.1057 | 0.7177±0.0968 | 0.0030±0.0009 | 0.9817±0.0059 | 0.5502±0.1060 | 0.4780±0.0950 |
| TEdit-CSDI | **0.1557**±0.0195 | **0.3009**±0.0266 | 0.7329±0.0654 | 0.7392±0.0639 | 0.0022±0.0004 | 0.9855±0.0020 | 0.5733±0.2235 | 0.5008±0.2151 |
| TEdit-TW | 0.1647±0.0202 | 0.3054±0.0187 | **0.7548**±0.0498 | **0.7658**±0.0343 | **0.0019**±0.0000 | **0.9869**±0.0010 | **0.7559**±0.0291 | **0.6813**±0.0307 |

|  | Overall | | Trend Type△ | | Trend Direction▲ | | Season Cycles▲ | |
|---|---|---|---|---|---|---|---|---|
|  | ↓MSE | ↓MAE | ↓\|RaTS\| | ↑CTAP | ↑RaTS | ↑CTAP | ↑RaTS | ↑CTAP |
| CSDI | 0.2396±0.0781 | 0.3780±0.0642 | 0.3819±0.1058 | 0.5349±0.1373 | 1.1210±0.0069 | 0.9788±0.0102 | 0.3091±0.0893 | 0.1987±0.0911 |
| Time Weaver | 0.2138±0.0191 | 0.3589±0.0146 | 0.2397±0.0787 | 0.7190±0.0972 | 1.1247±0.0002 | 0.9854±0.0007 | 0.5227±0.1156 | 0.4026±0.1005 |
| TEdit-CSDI | **0.1991**±0.0122 | 0.3461±0.0186 | 0.2255±0.0510 | 0.7317±0.0704 | 1.1248±0.0006 | 0.9850±0.0024 | 0.5588±0.2143 | 0.4385±0.2081 |
| TEdit-TW | 0.1998±0.0191 | **0.3411**±0.0125 | **0.2208**±0.0293 | **0.7626**±0.0217 | **1.1249**±0.0000 | **0.9859**±0.0009 | **0.7859**±0.0465 | **0.6610**±0.0491 |

Table 8:  Performance on all subsets of attributes combination of Synthetic, ▲/△ denote edited/preserved attributes.

|  | Air | | | | Motor | | | |
|---|---|---|---|---|---|---|---|---|
|  | City▲ | | Season△ | | Channel Id▲ | | Imagined Movement△ | |
|  | ↑RaTS | ↑CTAP | ↓\|RaTS\| | ↑CTAP | ↑RaTS | ↑CTAP | ↓\|RaTS\| | ↑CTAP |
| CSDI | 1.2878±0.0590 | 0.3287±0.0779 | 0.2349±0.0131 | 0.4473±0.0296 | 0.1895±0.0030 | 0.8395±0.0025 | 0.2963±0.0045 | 0.4402±0.0045 |
| Time Weaver | 1.4429±0.0709 | 0.5227±0.0861 | 0.2143±0.0074 | 0.4736±0.0440 | 0.1802±0.0085 | 0.8316±0.0061 | **0.2799**±0.0018 | **0.4542**±0.0100 |
| TEdit-CSDI | 1.3036±0.0565 | 0.3541±0.0664 | **0.2116**±0.0080 | 0.4807±0.0212 | 0.1925±0.0018 | 0.8418±0.0015 | 0.2927±0.0040 | 0.4467±0.0027 |
| TEdit-TW | **1.4602**±0.0556 | **0.5337**±0.0657 | 0.2226±0.0068 | **0.4839**±0.0095 | **0.1945**±0.0031 | **0.8443**±0.0020 | 0.2885±0.0094 | 0.4394±0.0113 |
|  | City△ | | Season▲ | | Channel Id△ | | Imagined Movement▲ | |
|  | ↓\|RaTS\| | ↑CTAP | ↑RaTS | ↑CTAP | ↓\|RaTS\| | ↑CTAP | ↑RaTS | ↑CTAP |
| CSDI | **0.1062**±0.0217 | 0.8149±0.0196 | 0.2026±0.0124 | -0.0124±0.0163 | **0.0231**±0.0014 | 0.8832±0.0029 | -0.0018±0.0124 | 0.0010±0.0059 |
| Time Weaver | 0.1588±0.0069 | 0.7863±0.0107 | 0.3483±0.0230 | 0.1304±0.0206 | 0.0241±0.0022 | 0.8840±0.0007 | 0.0153±0.0127 | 0.0021±0.0034 |
| TEdit-CSDI | 0.1112±0.0058 | **0.8250**±0.0099 | 0.3007±0.0245 | 0.0818±0.0217 | 0.0233±0.0008 | 0.8841±0.0031 | 0.0106±0.0022 | -0.0047±0.0083 |
| TEdit-TW | 0.1606±0.0414 | 0.7709±0.0458 | **0.4720**±0.0572 | **0.2523**±0.0521 | 0.0258±0.0010 | **0.8847**±0.0005 | **0.0479**±0.0069 | **0.0252**±0.0067 |

Table 9:  Performance on all subsets of attributes combination of Motor and Air, ▲/△ denote edited/preserved attributes.

## I.2 Sensitivity Study for multi-resolution

In this section, we show the impact of different Multi-resolution parameters on different attributes in Fig. 11. All experiments are performed on 6 subsets of Synthetic, including editing [trend type, trend direction, season cycles, trend type & trend direction, trend type & season cycles, trend direction & season cycles]. To analyze the influence of $R$, we fix $L_p = 2$ and vary $R$. To examine the impact of $L_p$, we fix $R = 3$ and vary $L_p$. We found that different attributes exhibit varying preferences for multi-resolution parameters across different subsets. This observation supports the motivation for proposing a multi-resolution approach: distinct attributes exert different scales of influence on time series data.

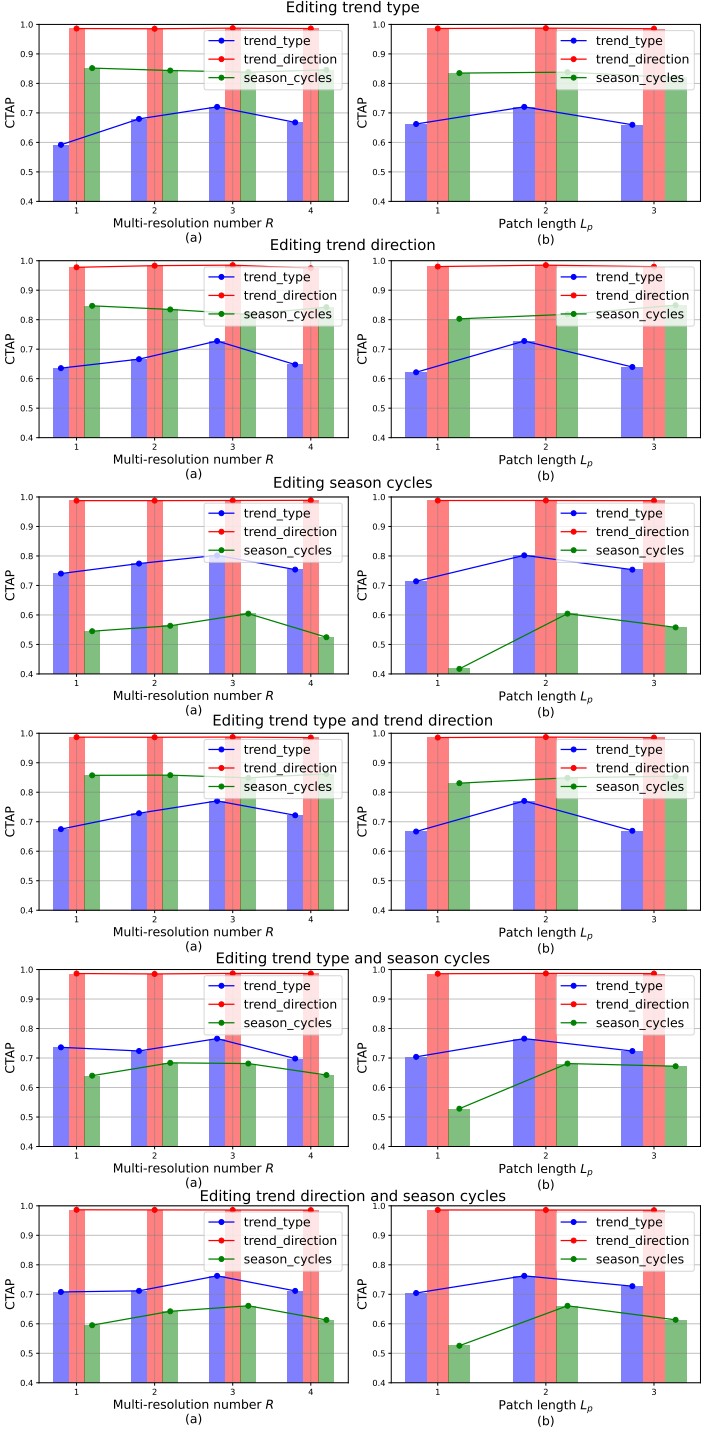

Figure 11: The impact of multi-resolution on 6 Synthetic subsets.

## I.3 Case Study

In this section, we demonstrate the superiority of our approach by visualizing the results of our method with a baseline on editing tasks. As shown in Fig. 12, we compare our method and baselines on Synthetic dataset. The left column compares the TEdit-CSDI and CSDI while the right column compares the TEdit-TW and Time Weaver. Each row represents different attribute settings, containing: modifying trend type, modifying trend direction, modifying season cycles, modifying trend type and trend direction, modifying trend type and season cycles, modifying trend direction and season cycles. It's obvious that the editing results of our method are much closer to the target time series with lower MSE and MAE, proving the superiority of our method.

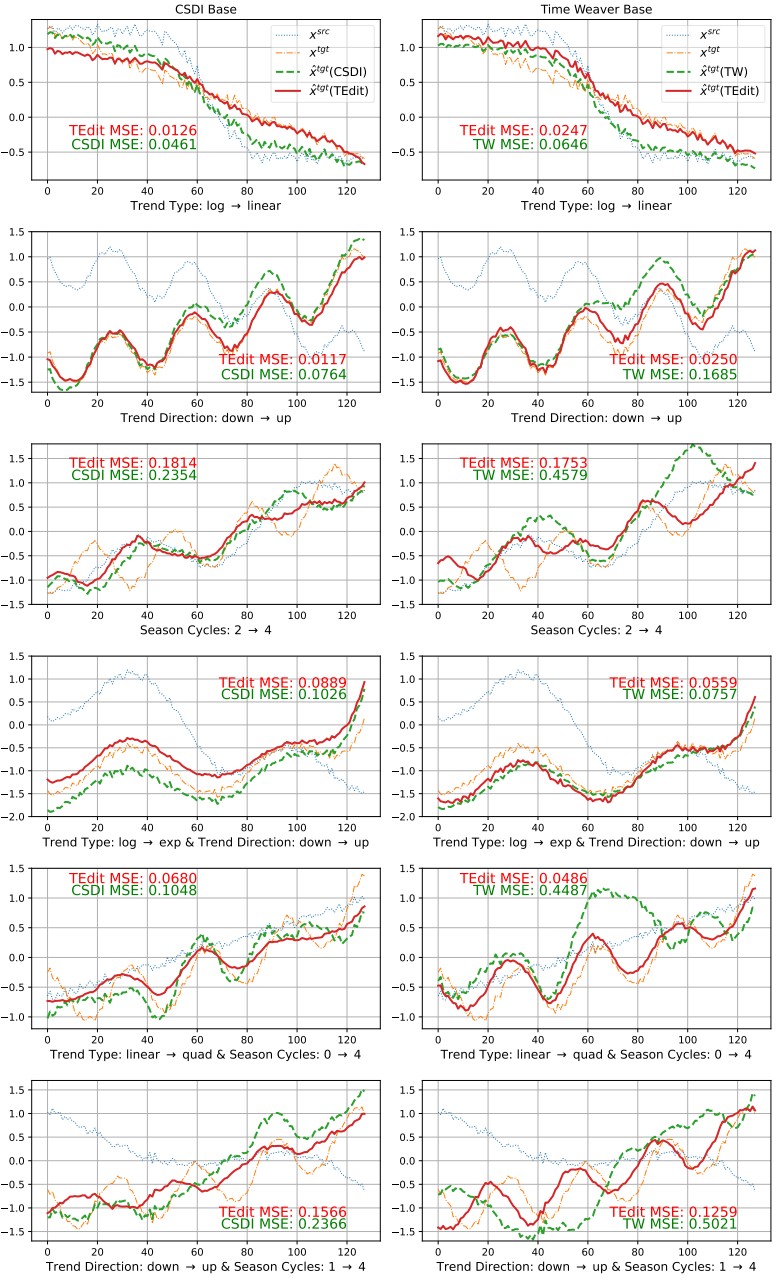

Figure 12: More cases of different attribute subsets of Synthetic dataset

## I.4 Comparing Editing and Conditional Generation

Here we provide more comparison results of conditional generation and editing in Fig. 13. The editing results are closer to the target time series with lower MSE and MAE. Although the time series generated by condition generation also meets the requirement of the attributes, it still lost the detailed information of input while the editing process better retains the characteristics of the time series to preserve and manipulate only the target attribute to change. Therefore, the editing can achieve a more precise control compared with conditional generation.

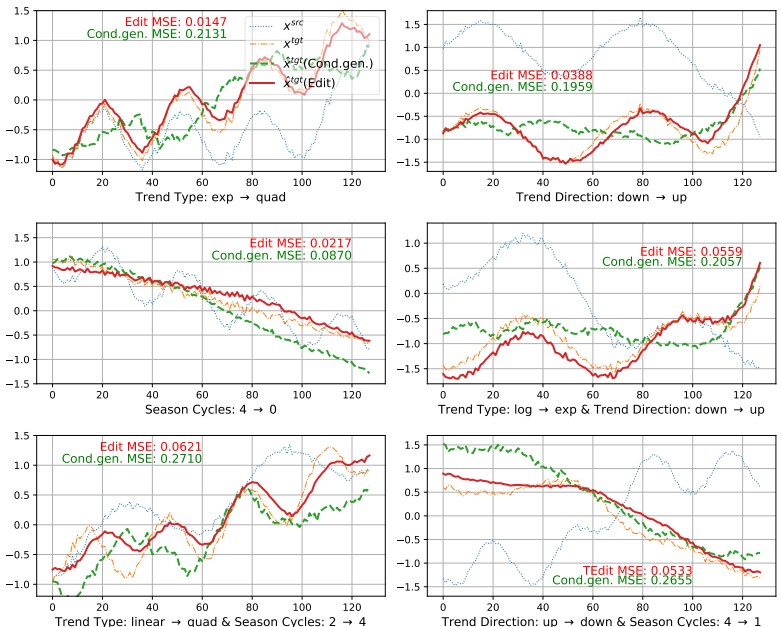

Figure 13: Comparison between editing and conditional generation

## I.5 Bootstrap influence on different attributes

To further explore the impact of bootstrapping on data coverage, we visualize the effect of bootstrapping data on different attributes to complement the original data. Fig. 14,15,16 present the results on trend types, and season cycles, trend directions, respectively. According to the classification accuracy of the TAP model and case study, we can infer that trend directions are a relatively simple pattern for modeling while the trend types and season cycles are more complex. The data coverage improvement brought by bootstrapping is more obvious when the attribute is complex.

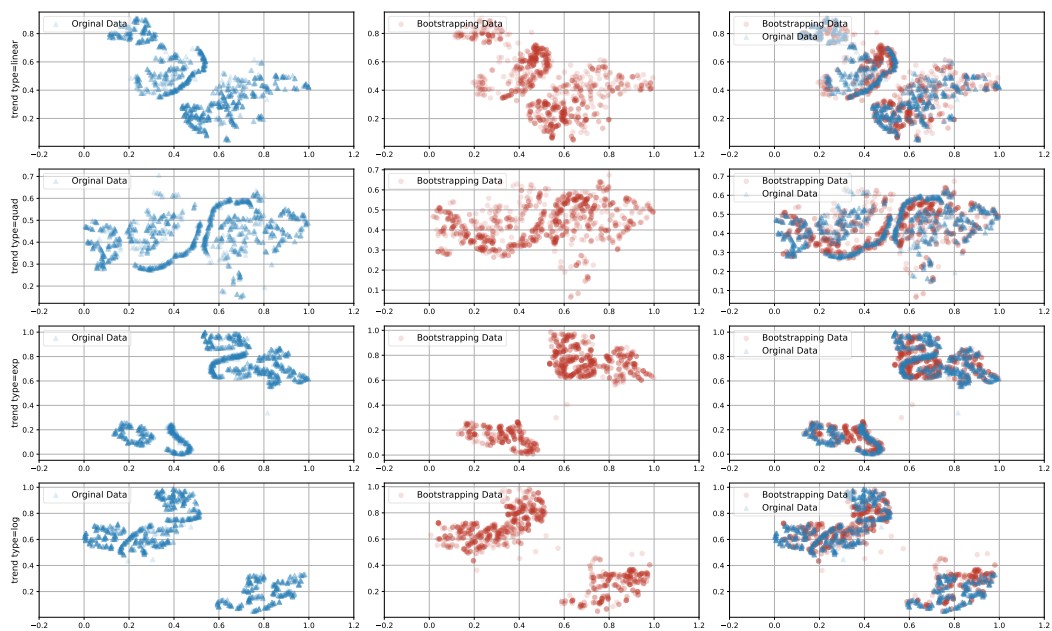

Figure 14: The influence of bootstrapping on different trend types

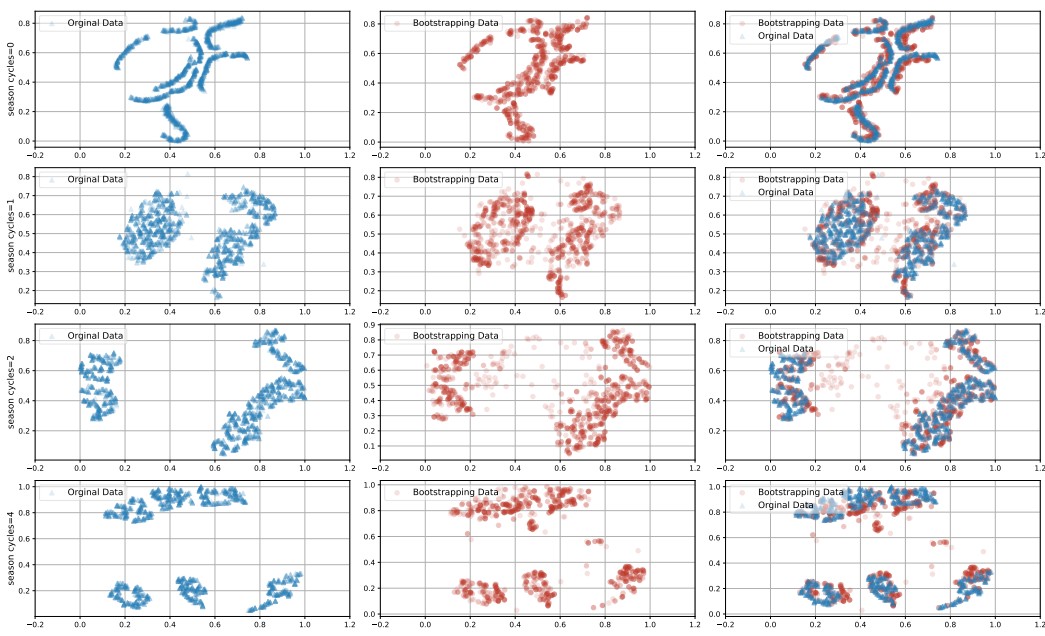

Figure 15: The influence of bootstrapping on different season cycles

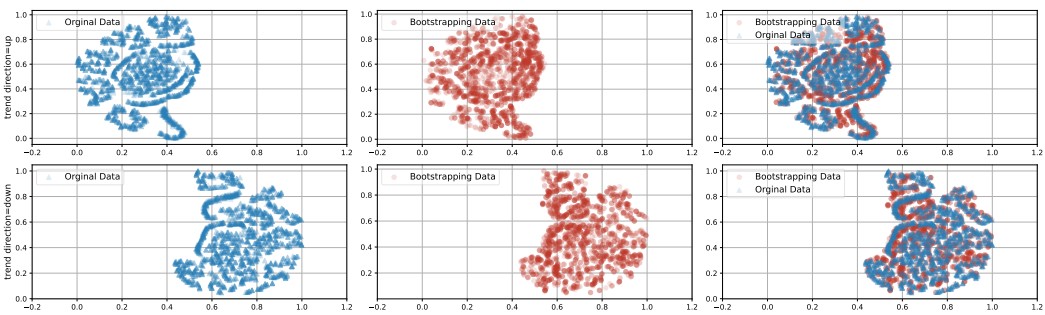

Figure 16: The influence of bootstrapping on different trend directions

