# OpenReview forum: "Towards Editing Time Series"
_NeurIPS.cc/2024/Conference — NeurIPS 2024 poster_

### Official Review · Reviewer_FqWw · 2024-07-04

**Soundness:** 2
**Presentation:** 3
**Contribution:** 3
**Rating:** 5
**Confidence:** 4

**Summary:**

This paper introduces Time Series Editing (TSE), a method for generating time series data with control over their attributes. TSE generates a sample by modifying an existing time series through manipulation of specific attributes while maintaining consistency in others. This approach leverages the hypothesis that patterns in time series synthesis are transferable between attribute values with consistent shared attributes. The authors propose a novel multi-resolution diffusion-based model for sample generation, that is trained using a bootstrapped learning algorithm. They show that this architecture and training procedure choice overcomes challenges related to biased data distribution and varying degrees of influence different attributes have on time series generation.

**Strengths:**

$\textbf{Importance of the task}$: I think the proposed method is tackling a very important problem that has potential in many applications. In fact, I would suggest that the authors highlight the potential of their approach even more. Generating such time series samples has many applications in different domains. Right now the paper focuses on the 'editing' aspect of TSE, but the proposed method has potential to generate counterfactual time series with control over attributes.

$\textbf{Multi-resolution generative model}$: The proposed multi-resolution diffusion model is an interesting design to overcome the varying scale of impact in time series.

**Weaknesses:**

$\textbf{Edited attribute set}$: In the experiment section, the results are for a specific type of change (a specific set of edited attributes). It is not clear why this specific combination was selected. I think the performance should be reported for all possible combinations, or the average of all possible combinations. Another interesting extension to this would be to look into these averaged performances, as a function of the size of the manipulated set of attributes. This might not be helpful for the selected real-world datasets because they only have 2 attributes, but at least the authors can assess this in the synthetic setting.

$\textbf{Standard errors}$: Some of the reported performances are very close and it is difficult to assess their significance. The authors need to add confidence intervals to the reported numbers in all tables.

$\textbf{RaTS}$: It is not clear how $p(a_k^{tgt}|x)$ is estimated in order to calculate the RaTS performance score?

$\textbf{MSE for real data}$: In general, it is difficult to know the ground-truth for edited time series in real-world dataset. However, I believe in the existing datasets, this is not impossible. For instance, in the Air dataset, if the changed attribute is the city, we have the corresponding time series in the data. This way the authors can also report MSE/MAE for the real-world datasets as well. This is important because even though the other metrics assess important qualities for the model, they cannot tell how well or realistic the generated signal is.  Adding this evaluation can help better assess the performance of TSE on real-data.

**Questions:**

$\textbf{Correlated attributes}$: In many real-world applications, the underlying attributes are not independent. As a result, randomly changing attributes while fixing others can result in out of distribution or unrealistic samples. I'm wondering if this becomes a problem especially in the bootstrap self-scoring step where the model will end up overfitting to out of distribution samples.

$\textbf{Distributional coverage}$: I'm not sure the conclusion drawn from Figure 5 is correct. The figure shows T-SNE projection of the data and the bootstrapped data, and the conclusion is that the generated samples fill in the gaps in the original data space. However, with methods like T-SNE, the projection is a function of the data they are trained on. Which means comparing the three subfigures that are representing different data cohorts is not possible. Can the authors provide more explanation on this?

$\textbf{Unseen attributes}$: Does the approach extend to unseen attribute values? Can you generate a time series with an edited attribute where the value has never been seen in the training data?

Note: I would be willing to increase my score if some of the concerns mentioned in the review are addressed.

**Limitations:**

$\textbf{Evaluating generated time series samples}$: One of the biggest challenges for generative models in time series is evaluating the generated samples. The proposed metrics and evaluations are useful in assessing different aspects of the performance of TSE, but there is no discussion on how we can assess whether a generated sample is realistic or not. Even a low MSE by itself doesn't guarantee high-quality generated samples. I think this is something worth discussing in the paper.

---

> ### Author Rebuttal · Authors · 2024-08-07
>
> Thanks for your constructive feedback!
>
> >**W1. Edited attribute set**
>
> Thanks for your comments.
>
> In the tables below, we show the average experimental results of editing different types and numbers of attributes on the Synthetic dataset. It can be seen that our TEdit outperforms baseline method on all metrics. The subscripts "edit" and "prsv" refer to the edited and preserved attributes. For detailed results of each combination, please refer to the PDF in the General Response. We will add more results in the revised version.
>
> - ### Table 1: averaged results of different edit-preserve attribute groups on Synthetic.
> |Method|MSE$\downarrow$|MAE$\downarrow$|RaTS_edit$\uparrow$|RaTS_abs_prsv$\downarrow$|
> |-|-|-|-|-|
> |Time Weaver|0.1502|0.2953|0.8537|0.1206|
> |TEdit-TW|**0.1184**|**0.2641**|**0.9599**|**0.1069**|
>
> >**W2. Standard errors**
>
> For each setting, we run the experiment three times, and calculate the mean and standard deviation (STD). These results can be seen in the PDF file in the General Response.
> From the results, the improvement of our method against the baseline is significantly larger than the STD values in almost all the settings, which illustrates that our method's improvements are significant. Moreover, the STD values of our method is much smaller than the baseline, showing that our method has better stability.
>
> >**W3. RaTS**
>
> Briefly speaking, we follow [1] and leverage an external model TAP (Time series - Attribute Pretraining) similar to CLIP [2] to estimate this probability. The details of TAP are presented in Appendix G.
>
> $p(a_{k}^{tgt}|x)$ is calculated in the following steps.
> - Use TAP to encode time series $\mathbf{x}$ and the $N_k$ possible values, e.g., linear and exponential, of the $k$-th attribute $a_k$, e.g., trend type, into embeddings $\mathbf{h}_{x}$ , and $\mathbf{h}\_{a_k}^{n}$, where $n\in[1,..., N_k]$.
> - Obtain $p(a_k^n|x)$ by applying a softmax over the cosine similarity between $\mathbf{h}_{x}$ and $\mathbf{h}\_{a_k}^{n}$.
>
> >**W4. MSE for real data**
>
> MSE is indeed a straightforward metric for measuring the quality of generated samples. However, MSE can be used only when source and target samples share **all** the attributes, including hidden attributes such as noise, except for the edited attributes. We can guarantee such a 1-1 pair for synthetic data. However, in real-world datasets, there are many unseen attributes and interference which make the time series vary a lot even with the same attribute combination. For example, air quality is not only influenced by city and season, but also other factors such as weather, wind and the industrialization level. We may not say that the edited time series by modifying city attribute from "Boston" to "New York" is exactly the same as the specific data sample with city of New York in the dataset.
>
> For the real-world datasets, we follow the practices in computer vision domain [1], by utilizing external models to evaluate the quality of the edited samples. We acknowledge that evaluating generative models is challenging, especially for the newly proposed time series editing task. We will keep investigating better benchmarking solutions in this novel direction.
>
> >**Q1. Correlated attributes**
>
> We also had similar concerns at the early stage of experiments. However, after statistically visualizing the data generated by bootstrapping, we found that the data generated by bootstrapping is a more complete complement to the original real data distribution, as shown in Fig. 5 in the paper. Training on both the real and the bootstrapped data makes the model fit the complete distribution and derive better performance, as illustrated by our experiments. Therefore, there is almost no overfitting outside the real distribution.
>
> >**Q2. Distributional coverage**
>
> Thanks for your comments and sorry for the misleading description.
>
> We use all the data, including both real and bootstrapped data, for T-SNE. Thus, the three subfigures of Fig. 5 share the same T-SNE model with the same projection, which means their data cohorts are the same. So it is reasonable to comapre the distribution coverage of the original real dataset, bootstrapped synthetic dataset, and the full dataset of the above two sets, in the three subfigures. We will further clarify this in the revised version.
>
> >**Q3. Unseen attributes**
>
> Our understanding of the "unseen attribute values" is "unseen attribute combinations". In other words, the model has seen all $N_k$ values of each individual attribute $a_k$, but has not seen certain attribute combinations. This setting is the same as our synthetic datasets. There are 4 unseen attribute combinations in the test data. Therefore, our proposed TEdit could deal with the unseen attribute combinations. In the future, we plan to further explore this interesting setting for real-world datasets.
>
> >**L1. Evaluating generated time series samples**
>
> Thanks for your insightful comments!
>
> Assessing the reality of the generated data has always been a critical open challenge. This is extremely challenging even for images, where attributes such as color and size can be easily interpreted, not to mention time series, where the data is difficult to be interpreted. The classic work [3] asked humans to evaluate the reality of the generated images. We follow the practice of recent works [1] to use external models as evaluation. Nevertheless, assessing the reality can be an interesting future direction. We will further discuss it in the revised version.
>
> [1] Li et al. Blip-diffusion: Pre-trained subject representation for controllable text-to-image generation and editing. NeurIPS'2023.
>
> [2] Radford et al. Learning transferable visual models from natural language supervision. ICML'2021.
>
> [3] Meng et al. SDEdit: Guided Image Synthesis and Editing with Stochastic Differential Equations. ICLR'2022

---

> > ### Comment · Reviewer_FqWw · 2024-08-10
> > **Rebuttal response**
> >
> > Thank you for the detailed rebuttal. I believe the added evaluations and results will be a good addition to the paper. The rebuttal has covered most my concerns and I'm happy to edit my score.

---

> > > ### Author Response · Authors · 2024-08-13
> > >
> > > Thank you so much for your recognition and we are very much encouraged by your positive feedback and increased scores.
> > >
> > > We will further improve our paper according to your suggestions. If there are any other suggestions or questions you'd like to discuss, please don't hesitate to let us know. We are happy to further discuss the opportunities and challenges of this new and exciting Time Series Editing (TSE) task.

---

### Official Review · Reviewer_EFAk · 2024-07-05

**Soundness:** 3
**Presentation:** 3
**Contribution:** 2
**Rating:** 7
**Confidence:** 4

**Summary:**

The paper discusses the challenges of synthesizing time series data influenced by intrinsic and extrinsic factors. It highlights the limitations of existing benchmarks and methods and suggests a new task, Time Series Editing (TSE), which modifies given time series based on specified properties while preserving other attributes. In addition, the paper suggests a new method based on generative bootstrapping and the incorporation of multi-resolution modeling to enhance data coverage, demonstrating its efficacy in experiments.

**Strengths:**

- The task of editing specific attributes is novel and very interesting.
- Providing end-to-end benchmarks that include datasets,  metrics, and baselines
- Suggesting  a new baseline method based on bootstrapping and multiresolution

**Weaknesses:**

- The work "On the Constrained Time-Series Generation Problem" [1] introduces time series generation and editing. Although there are certain differences between the current work and the above work, they can be seen as similar. However, the paper does not reference or discuss the similarities and differences. I think that incorporating such a comparison is crucial. In addition, this somehow negatively affects the first novelty that the offers suggested. Nevertheless, I still think the task is novel enough.

- The main experiment results in Tables 1 and 2 are on specific attributes; to show robustness, it is necessary to show or average all the possible preserved-changed attribute groups.

Multi-resolution models for time series tasks are already in use [2, 3], which limits the novelty. I want to caveat this weakness by stating that [2] is not in the context of generation, and [3] is almost concurrent with this work.


[1] - On the constrained time-series generation problem. Coletta, Andrea and Gopalakrishnan, Sriram and Borrajo, Daniel and Vyetrenko, Svitlana.
[2] - Multi-resolution Time-Series Transformer for Long-term Forecasting. Yitian Zhang, Liheng Ma, Soumyasundar Pal, Yingxue Zhang, Mark Coates
[3] - Multi-Resolution Diffusion Models for Time Series Forecasting. Lifeng Shen, Weiyu Chen, James Kwok

**Questions:**

- Can the author present all the possible preserved-edited groups in the main results? Currently, in my opinion, the main results are not extensive enough.

- Why did the author not mention "On the Constrained Time-Series Generation Problem"?

**Limitations:**

detailed on the weakness

---

> ### Author Rebuttal · Authors · 2024-08-07
>
> Thanks for your valuable comments!
>
> >**W1. Comparison with [1]**
>
> Thanks for pointing it out, and we will include a discussion and comparison of [1] in the revised version. Time Series Editing (TSE) proposed in our work is essentially different to [1], though these two works share some similarities in conditional generation.
>
> Specifically, the term "constraint" mentioned in [1] is actually similar to the term "condition" or "attribute" used in our work and [2]. The task of [1] can be formulated as $\mathbf{x}^{tgt}=\Phi(\mathbf{a}^{tgt})$, where $\mathbf{a}^{tgt}$ is the target constraint, $\mathbf{x}^{tgt}$ is the desired output and $\Phi$ denotes the function of generative model to be learned. However, our proposed TSE is formulated as $\mathbf{x}^{tgt}=\Phi(\mathbf{x}^{src}, \mathbf{a}^{src}|\mathbf{a}^{tgt})$, which generates $\mathbf{x}^{tgt}$ based on the input source sample $(\mathbf{x}^{src}, \mathbf{a}^{src})$ and the target condition $\mathbf{a}^{tgt}$. Since [1] and [2] are both "conditional generation" studies, we compare with the more recent work [2] in our current version. We will discuss and compare with [1] in the revised verison.
>
> >**W2. Averaged results**
>
> Thanks for your feedback.
>
> Below we provide the experimental results of the average performance of editing different attributes for reference, hoping to help you better understand the performance of different methods. We also provide the results for all 6 edit-preserve attribute groups in the pdf of the general response.
>
> In the following tabels we present the avergaed results of editing different attribute combinations on the Synthetic and Air datasets. The Synthetic dataset has 3 attribtues: trend type, trend direction and season cycle. The Air dataset has 2 attributes: city and season.
> The results show that our proposed method TEdit outperforms the baseline on all the evaluation metrics.
>
> - ### Table 1. Average performance of editing different attributes on Synthetic.
> |Method|MSE$\downarrow$|MAE$\downarrow$|RaTS_edited$\uparrow$|RaTS_abs_preserved$\downarrow$|
> |:--------:|:---------:|:--------:|:---------:|:--------:|
> |Time Weaver|0.1502|0.2953|0.8537|0.1206|
> |TEdit-TW|**0.1184**|**0.2641**|**0.9599**|**0.1069**|
>
> - ### Table 2. Average performance of editing different attributes on Air.
> |Method|RaTS_edited$\uparrow$|RaTS_preserved$\downarrow$|
> |:--------:|:--------:|:--------:|
> |Time Weaver|0.8799|0.2017|
> |TEdit-TW|**0.9970**|**0.1753**|
>
> >**W3. Multi-resolution**
> Thank you for sharing these works, it will actually enhance the discussion of related works of our paper.
>
> As mentioned by your comments, one work is almost concurrent with our paper and these works primarily focus on the forecasting task, aiming to make more accurate predictions.
> Our work, however, focuses on the editing task, which requires to generate the corresponding time series regarding the given editing attributes as condition. The reason we propose the multi-resolution approach is that different attributes impact the time series at different scales. For example, the trend type has a global impact, while the season number focuses on more local details. In our work, we consider both the multi-resolution aspects of the time series and attributes, and leverage multi-resolution mechanism on both modeling and generation, which is more reflective of reality.
> Thank you again for mentioning these related works. We will discuss them in the revised version of our paper.
>
> [1] Coletta et al. On the constrained time-series generation problem. NeurIPS'2023
>
> [2] Narasimhan, Sai Shankar, et al. Time weaver: A conditional time series generation model. ICML'2024

---

> > ### Comment · Reviewer_EFAk · 2024-08-11
> > **Respose**
> >
> > Thank you for your response.
> >
> > Regarding the first concern, I agree that clarifying the differences in task definitions is important. In my opinion, contrasting these definitions clearly is crucial. Additionally, a universal formulation for both tasks should be considered. For instance, if x_source  and a_source are left empty, both tasks could be viewed as the same. Therefore, I believe that these problems should be unified under a single benchmark. I encourage the authors to explore how this framework can be unified in the final revision of the paper.
> >
> > As for my second concern, I am unclear if the reported results are averaged across all editing scenarios. The title mentions "different attributes," which suggests that it may only cover a subset of the possible editing scenarios. I believe an extensive evaluation across all possible editing scenarios should be conducted on both datasets to thoroughly assess the method's robustness. Could you clarify this?
> >
> > Lastly, thank you for providing clarification on the third concern. I believe the approach you've provided is solid enough to serve as a reliable baseline for benchmarking, which is significant. As for the claimed novelty, I think it should be specifically attributed to adapting the multiresolution approach for time series editing and generation.

---

> > > ### Author Response · Authors · 2024-08-13
> > >
> > > Thank you so much for your insightful comments!
> > >
> > > For the first concern, indeed, leaving $\mathbf{x}^{src}$ and $\mathbf{a}^{src}$ empty, the editing task will degenerate to the conditional generation task, as is mentioned by your comments.
> > > Therefore, *conditional generation could be regarded as a special case of editing*.
> > > Actually, we can directly evaluate the performance of conditional generation on our editing datasets by masking out the information of the source $\mathbf{x}^{src}$ and $\mathbf{a}^{src}$.
> > > Specifically, for the editing task, each sample has three inputs $\mathbf{x}^{src}$, $\mathbf{a}^{src}$ and $\mathbf{a}^{tgt}$; for the conditional generation task, only $\mathbf{a}^{tgt}$ is available to models.
> > >
> > > To further enhance our experiments regarding to your suggestions, we conducted additional experiments on the conditional generation task, which will be put into the revised paper version as a supplementary experimental setting of our work.
> > > In the table below, we report the averaged performance of conditional generation of Time Weaver and our TEdit-TW on different Synthetic datasets: all 6 possible edit-preserve attribute groups.
> > > Please note that the RaTS score is not suitable for the conditional generation and we have not reported it in this new experiment,
> > > since RaTS scores are calculated based on both $\mathbf{a}^{src}$ and $\mathbf{a}^{tgt}$, and $\mathbf{a}^{src}$ is unavailable for the conditional generation task.
> > > It can be seen that our method TEdit-TW can significantly outperform Time Weaver for the conditional generation task.
> > > Following your suggestions, we will add the conditional generation task to our benchmark in the revised version. Thank you for your valuable suggestion.
> > >
> > > #### Tabel 1. Averaged performance of conditional generation on 6 different Synthetic datasets: all 6 possible edit-preserve attribute groups.
> > > |Method|MSE$\downarrow$|MAE$\downarrow$|TAP_trend_type$\uparrow$|TAP_trend_direction$\uparrow$|TAP_season_cycle$\uparrow$|
> > > |:--------:|:--------:|:--------:|:--------:|:---------:|:--------:|
> > > |Time Weaver|0.3509|0.4691|0.7685|0.9880|0.2035|
> > > |TEdit-TW|0.2414|0.3958|0.8353|0.9899|0.2568|
> > >
> > >
> > > Regarding the second concern, we are sorry that the "different attributes" in the title caused the confusion.
> > > The "different attributes" actually means "all possible combinations of edit-preserve groups".
> > > For the synthetic dataset, it includes the following (edit | preserve) sets:
> > >
> > > - (trend type | trend direction, season cycle)
> > > - (trend direction | trend type, season cycle)
> > > - (season cycle | trend type, trend direction)
> > > - (trend type, trend direction | season cycle)
> > > - (trend type, season cycle | trend direction)
> > > - (trend direction | trend type, season cycle).
> > >
> > > For the Air dataset, it includes the following (edit | preserve) sets:
> > > - (city | season)
> > > - (season | city)
> > >
> > > The scores in Tables 1 and 2 in the previous response are averaged over all these combinations, and the scores for each edit-preserve group are presented in the PDF files of the general response.
> > >
> > > As for the third concern, as is mentioned by your comments, our novelty lies in the multiresolution modeling in time series editing and generation task, which is unique and different compared to the concurrent related works mentioned by your previous review comments.
> > > We are encouraged by your positive comments and we think that a more proper description would help better understand these works in the revised paper.
> > >
> > > Thank you once again for acknowledging our work. We will carefully revise our paper in line with your valuable suggestions, including:
> > > * Unifying the definition framework of editing with conditional generation and providing a clearer discussion of their differences and distinctions.
> > > * Incorporating additional experiments on our method within the conditional generation setting, to enhance the comprehensiveness of our experiments.
> > > * Refining our claims to more properly describe our contributions, particularly in relation to the multiresolution mechanism in generation and editing tasks.
> > >
> > > We are happy to have more discussions if you have any other concerns or suggestions.

---

> > > > ### Comment · Reviewer_EFAk · 2024-08-13
> > > > **Response**
> > > >
> > > > Thank you. You have addressed all of my concerns. After the rebuttal and the reviewer's comments, I think the paper will be more substantial and impactful with the revised version. Therefore, I will raise my score.
> > > >
> > > > If I may add a comment, I find this task very interesting. I hope that more works that will add real-world datasets or complex synthetic data will come in the future and challenge this task, as today is pretty simple. Thanks.

---

> > > > > ### Author Response · Authors · 2024-08-13
> > > > >
> > > > > Thank you!
> > > > >
> > > > > We are happy to hear that all the concerns have been addressed!
> > > > > We are pretty much encouraged by your constructive and insightful comments and feedback!
> > > > > We will further revise our paper according to your suggestions.

---

### Official Review · Reviewer_ZaGr · 2024-07-11

**Soundness:** 3
**Presentation:** 4
**Contribution:** 4
**Rating:** 6
**Confidence:** 4

**Summary:**

This paper introduces time series editing (TSE), a novel approach for modifying existing time series to match specified attributes while preserving other attributes. Unlike traditional methods that generate time series from scratch, TSE employs a multi-resolution modeling and bootstrap tuning framework, enhancing data coverage and generation precision. The paper presents a multi-resolution diffusion model and a comprehensive benchmark dataset, demonstrating the method’s effectiveness on both synthetic and real-world data.

**Strengths:**

- The paper introduces the novel concept of Time Series Editing (TSE), a significant innovation in the field of time series data synthesis.
- This work offers a practical solution for generating high-quality time series data with specified attributes, addressing data sparsity and enhancing the utility of time series synthesis. The novel approach and its demonstrated effectiveness on diverse datasets highlight its broad applicability and potential to drive future research and applications in the field.
- The overall quality of the paper is exemplary, demonstrated by thorough experimentation and rigorous evaluation. The authors present a well-designed DDIM model with the proposed multi-resolution noise estimator and bootstrapped training algorithm.
- The paper is clearly written and well-structured, making it accessible to readers with varying levels of expertise in time series analysis.

**Weaknesses:**

- The paper focuses on editing specific attributes of time series while preserving others but does not extensively discuss the handling of diverse attribute types and their interactions.
- The reproducibility is uncertain without code and detailed implementation information in Appendix I.
- There are minor writing issues, such as “levela” in line 291, page 8, that need correction.

**Questions:**

I have no specific questions.

**Limitations:**

The authors adequately addressed the limitations.

---

> ### Author Rebuttal · Authors · 2024-08-07
>
> Thanks for your appreciation and feedbacks! Below are the responses.
>
> >**W1. Handling of diverse attribute types and their interactions.**
>
> Thanks for your comments.
>
> The following tables 1~6 show the complete experimental results of editing different combination of attributes (trend type, trend direction and the number of season cycles), hoping to provide a better understanding of our method.
> Due to time limitation, in the following tables, we present the results of TEdit-TW and Time Weaver [1] on editing different sets of attributes on Synthetic dataset.
> From the tables, our TEdit-TW outperforms Time Weaver [1] on all attribute combinations.
> We also found that the difficulty of editing different attributes varies greatly. By comparing MSE, (1) editing season_cycle is the most difficult, editing trend_type is the easiest, and (2) editing multiple attributes is usually more difficult than a single one.
> For more detailed experimental results, please refer to the PDF file in the General Response.
>
> - ### Table 1. Synthetic dataset: edit "trend type".
> |Method|MSE$\downarrow$|MAE$\downarrow$|RaTS:trend_type$\uparrow$|TAP:trend_type$\uparrow$|RaTS_abs:trend_direction$\downarrow$|TAP:trend_direction$\uparrow$|RaTS_abs:season_cycle$\downarrow$|TAP:season_cycle$\uparrow$|
> |-|-|-|-|-|-|-|-|-|
> |Time Weaver|0.0565/0.0115|0.1875/0.0229|0.7532/0.0656|0.5940/0.0686|0.0017/0.0002|0.9866/0.0012|0.0913/0.0245|0.8424/0.0132|
> |TEdit-TW|0.0431/0.0008|0.1629/0.0015|0.8139/0.0093|0.6621/0.0052|0.0016/0.0001|0.9879/0.0003|0.0818/0.0010|0.8582/0.0010|
>
> - ### Table 2. Synthetic dataset: edit "trend direction".
> |Method|MSE$\downarrow$|MAE$\downarrow$|RaTS_abs_trend_type$\downarrow$|TAP_trend_type$\uparrow$|RaTS_trend_direction$\uparrow$|TAP_trend_direction$\uparrow$|RaTS_abs_season_cycle$\downarrow$|TAP_season_cycle$\uparrow$|
> |-|-|-|-|-|-|-|-|-|
> |Time Weaver|0.1178/0.0181|0.2844/0.0214|0.4211/0.07058|0.5835/0.0739|1.9720/0.0010|0.9831/0.0044|0.0924/0.0249|0.8083/0.0394|
> |TEdit-TW|0.0965/0.0010|0.2636/0.0013|0.3426/0.0174|0.6656/0.0101|1.9716/0.0005|0.9826/0.0003|0.0747/0.0098|0.8282/0.0056|
>
> - ### Table 3. Synthetic dataset: edit "season cycle".
> |Method|MSE$\downarrow$|MAE$\downarrow$|RaTS_abs_trend_type$\downarrow$|TAP_trend_type$\uparrow$|RaTS_abs_trend_direction$\downarrow$|TAP_trend_direction$\uparrow$|RaTS_season_cycle$\uparrow$|TAP_season_cycle$\uparrow$|
> |-|-|-|-|-|-|-|-|-|
> |Time Weaver|0.2294/0.0130|0.3699/0.0090|0.1467/0.0299|0.6988/0.0092|0.0014/0.0001|0.9868/0.0007|0.4991/0.1420|0.2321/0.1344|
> |TEdit-TW|0.2004/0.0025|0.3608/0.0025|0.1471/0.0027|0.7768/0.0023|0.0012/0.0001|0.9887/0.0002|0.8086/0.0088|0.5230/0.01258|
>
> - ### Table 4. Synthetic dataset: edit "trend type" and "trend direction".
> |Method|MSE$\downarrow$|MAE$\downarrow$|RaTS_trend_type$\uparrow$|TAP_trend_type$\uparrow$|RaTS_trend_direction$\uparrow$|TAP_trend_direction$\uparrow$|RaTS_abs_season_cycle$\downarrow$|TAP_season_cycle$\uparrow$|
> |-|-|-|-|-|-|-|-|-|
> |Time  Weaver|0.0793/0.0091|0.2216/0.0122|0.6438/0.0539|0.6351/0.0567|1.1216/0.0008|0.9865/0.0011|0.0731/0.0114|0.8401/0.0197|
> |TEdit-TW|0.0504/0.0007|0.1743/0.0009|0.7194/0.0033|0.7070/0.0019|1.1217/0.0008|0.9871/0.0001|0.0575/0.0020|0.8641/0.0014|
>
> - ### Table 5. Synthetic dataset: edit "trend type" and "season cycle".
> |Method|MSE$\downarrow$|MAE$\downarrow$|RaTS_trend_type$\uparrow$|TAP_trend_type$\uparrow$|RaTS_abs_trend_direction$\downarrow$|TAP_trend_direction$\uparrow$|RaTS_season_cycle$\uparrow$|TAP_season_cycle$\uparrow$|
> |-|-|-|-|-|-|-|-|-|
> |Time Weaver|0.1873/0.0031|0.3310/0.0018|0.6564/0.0381|0.6721/0.0421|0.0022/0.0002|0.9844/0.0004|0.4523/0.1320|0.3880/0.1244|
> |TEdit-TW|0.1453/0.0001|0.2939/0.0005|0.6959/0.0078|0.7283/0.0045|0.0018/0.0001|0.9883/0.0001|0.6924/0.0105|0.6233/0.0080|
>
> - ### Table 6. Synthetic dataset edit: "trend direction" and "season cycle".
> |Method|MSE$\downarrow$|MAE$\downarrow$|RaTS_abs_trend_type$\downarrow$|TAP_trend_type$\uparrow$|RaTS_trend_direction$\uparrow$|TAP_trend_direction$\uparrow$|RaTS_season_cycle$\uparrow$|TAP_season_cycle$\uparrow$|
> |-|-|-|-|-|-|-|-|-|
> |Time Weaver|0.2307/0.0122|0.3776/0.0108|0.2556/0.0409|0.6810/0.0536|1.1252/0.0005|0.9862/0.0009|0.4601/0.1571|0.3448/0.1526|
> |TEdit-TW|0.1747/0.0011|0.3289/0.0013|0.2534/0.0080|0.7352/0.0075|1.1253/0.0003|0.9872/0.0002|0.6906/0.0059|0.5762/0.0044|
>
> >**W2. Reproducibility.**
>
> Thanks for your comments.
>
> We will release our codes upon the acceptance of this paper, and we will figure out a better way to present the implementation. In the current version of our paper, we have listed the implementation details and training details in the appendix. Specifically, Appendix I provides the training configurations. Appendix H shows the architectures of our model. Appendix E and F present evaluation model and metrics. Moreover, Appendix C and D illustrate the optimization details and training algorithm. Appendix E discusses the data processing details. Appendix F explains the evaluations. We will refine the discussion of the above implementation details for better reproducibility.
>
> >**W3. Writing suggestions.**
>
> Sorry for the inconvenience. We will make more rounds of proofreading and revise the details.
>
> [1] Narasimhan et al. Time weaver: A conditional time series generation model. ICML'2024

---

> > ### Comment · Reviewer_ZaGr · 2024-08-12
> > **Response from Reviewer ZaGr**
> >
> > Thank you for your detailed response, which addresses some of my concerns. I appreciate the effort you put into it. I will maintain my current rating unless more direct replication evidence is provided.

---

> > > ### Author Response · Authors · 2024-08-13
> > >
> > > Thanks for your positive feedback!
> > >
> > > We are happy to hear that most of your concerns have been addressed.
> > > As for the replication, we have sent AC an anonymous version of the code and data for the replication test, which is allowed under the regulation of NeurIPS rebuttal.
> > > Due to the rebuttal policy of NeurIPS, we are not allowed to add external links in the comments.
> > > We promise that we will publish the whole framework including the data preparation, training, and evaluation pipeline, for reproducibility upon the acceptance of our paper.
> > >
> > > Here, we provide the details of our implementation as below, including pseudo code for training and evaluation, model architectures, and hyper-parameters, hoping to better help improve the reproducibility of our work.
> > >
> > > ## 1. Pseudo code
> > > >**Pre-training**
> > > >
> > > >Inputs: \#epochs $N_{epoch}$, \#batches for each epoch $N_{batch}$, batch size $B$, noise estimator $\epsilon_\theta$, pretraining dataset $\mathcal{D}$, total diffusion step $T$, variance schedule $\\{\beta_t\\}_{t=1}^T$
> > > >
> > > >For epoch $n_{epoch}<N_{epoch}$:
> > > >
> > > >$\qquad$For batch $n_{batch}<N_{batch}$:
> > > >
> > > >$\qquad\qquad$ \# 1. Load data
> > > >
> > > >$\qquad\qquad$ Load a batch of $\mathbf{X}, \mathbf{A}$
> > > >
> > > >$\qquad\qquad\qquad$ \# 2. Train $\Phi$ via Eq.1
> > > >
> > > >$\qquad\qquad\qquad$ For each sample $\mathbf{x}\in\mathbf{X},\mathbf{a}\in\mathbf{A}$:
> > > >
> > > >$\qquad\qquad\qquad$ $t\sim Uniform(1,T)$ \# Sample a diffusion step $t$
> > > >
> > > >$\qquad\qquad\qquad$ $\epsilon\sim\mathcal{N}(0,1)$ \# Sample Gaussian noise $\epsilon$
> > > >
> > > >$\qquad\qquad\qquad$ $\mathbf{x}\_t=\sqrt{\alpha\_t}\mathbf{x}\_0+\sqrt{1-\alpha\_t}{\epsilon},(0,\mathbf{I}),~\alpha\_t:=\Pi\_{s=1}^t(1-\beta\_s)$ \# Get noisy time series $\mathbf{x}\_t$
> > > >
> > > >$\qquad\qquad\qquad$ $\hat{\epsilon}=\epsilon_\theta(\mathbf{x\_t},t,\mathbf{a})$ \# Predict the noise
> > > >
> > > >$\qquad\qquad\qquad$ $l=||\hat{\epsilon}- \epsilon||^2$ \# Noise estimation loss
> > > >
> > > >$\qquad\qquad\qquad$ Update the noise estimator $\epsilon_\theta$ by minimizing $l$
> > >
> > > >**Finetuning**
> > > >
> > > >Inputs: \#epochs $N_{epoch}$, \#batches for each epoch $N_{batch}$, batch size $B$, diffusion model $\Phi$ (the core component is the noise estimator $\epsilon_\theta$), bootstrap ratio $\psi$, finetuning dataset $\mathcal{D}$
> > > >
> > > >For epoch $n_{epoch}<N_{epoch}$:
> > > >
> > > >$\qquad$For batch $n_{batch}<N_{batch}$:
> > > >
> > > >$\qquad\qquad$ \# 1. Load data
> > > >
> > > >$\qquad\qquad$ Load a batch of $\mathbf{X}^{src}, \mathbf{A}^{src}, \mathbf{A}^{tgt}$
> > > >
> > > >$\qquad\qquad$ \# 2. Bootstrapping
> > > >
> > > >$\qquad\qquad$  $\hat{\mathbf{X}}^{tgt}\leftarrow\Phi(\mathbf{X}^{src},\mathbf{A}^{src}|\mathbf{A}^{tgt})$ \# Generate the target $\mathbf{X}^{tgt}$ by editing $\mathbf{X}^{src}$ as Eqs.4-5
> > > >
> > > >$\qquad\qquad$ $\hat{\mathbf{X}}^{src}\leftarrow\Phi(\hat{\mathbf{X}}^{tgt},\mathbf{A}^{src}|\mathbf{A}^{tgt})$ \# Back-translation Eqs.4-5
> > > >
> > > >$\qquad\qquad$ $\mathbf{s}\leftarrow MSE(\hat{\mathbf{X}}^{src}, \mathbf{X}^{src})$ \# Get self-score via MSE
> > > >
> > > >$\qquad\qquad$ $\hat{\mathbf{X}}^{tgt}\leftarrow Sort(\hat{\mathbf{X}}^{tgt}|\mathbf{s})$ \# Sort $\hat{\mathbf{X}}^{tgt}$ based on $\mathbf{s}$
> > > >
> > > >$\qquad\qquad$ $\hat{\mathbf{X}}^{tgt}\leftarrow\hat{\mathbf{X}}^{tgt}[:,B\cdot\psi]$ \# Keep the top $\psi$ samples with lowest MSE scores $\hat{\mathbf{x}}^{tgt}\in\hat{\mathbf{X}}^{tgt}$
> > > >
> > > >$\qquad\qquad$ \# 3. Calculate loss
> > > >
> > > >$\qquad\qquad$ Calculate the noise estimation loss based on the bootstrapped sample $\hat{\mathbf{X}}^{tgt}$. \# Similar to Eq.1 for pertaining
> > > >
> > > >$\qquad\qquad$ Update the model $\Phi$ via backpropogation.
> > >
> > > >**Evaluation**
> > > >
> > > >Inputs: model $\Phi$, dataset $\mathcal{D}$, \#batches $N_{batch}$, Time series Attribute Pretraining model (TAP)
> > > >
> > > >For batch $n_{batch}<N_{batch}$:
> > > >
> > > >$\qquad$ \# 1. Load data
> > > >
> > > >$\qquad$ Load a batch of $\mathbf{X}^{src}, \mathbf{A}^{src}, \mathbf{A}^{tgt}$, $\mathbf{X}^{tgt}$  \# $\mathbf{X}^{tgt}$ is only available for Synthetic data
> > > >
> > > >$\qquad$ \# 2. Generate $\hat{\mathbf{X}}^{tgt}$ by editing $\mathbf{X}^{src}$
> > > >
> > > >$\qquad$ $\hat{\mathbf{X}}^{tgt}\leftarrow\Phi(\mathbf{X}^{src},\mathbf{A}^{src}|\mathbf{A}^{tgt})$
> > > >
> > > >$\qquad$ \# 3. Evaluate $\hat{\mathbf{X}}^{tgt}$
> > > >
> > > >$\qquad$ Calculate RaTS scores (Appendix F.3) and TAP scores (Appendix F.2) based on the TAP model (Appendix G).
> > > >
> > > >$\qquad$ Calculate $MSE(\hat{\mathbf{X}}^{tgt}, \mathbf{X}^{tgt})$ if $\mathbf{X}^{tgt}$ is available

---

> > > > ### Author Response · Authors · 2024-08-13
> > > >
> > > > TAP is similar to the popular CLIP [2], and the purpose of TAP is to align time series $\mathbf{x}$ and attributes $\mathbf{a}$ in the embedding space, which is trained by distinguishing the positive and negative ($\mathbf{x}$,$\mathbf{a}$) pairs.
> > > > The details of TAP are presented in Appendix G, including architecture, pseudocode for training, and its performance.
> > > > TAP has two encoders: (1) a time series encoder, which is based on PatchTST [3] and (2) an attribute encoder (an MLP).
> > > > For different datasets, parameters are tuned e.g., patch lengths, 8,16,32 .., learning rate 1e-2, 1e-3, 1e-5 ...
> > > > On average, top1/top2 accuracy for different attributes of the Synthetic, Air and Motor datasets are 0.9145/0.9797, 0.9397/0.9746, 0.9258/0.9682 respectively.
> > > >
> > > > ## 2. Model Architectures
> > > > Our implementation of the noise estimator $\bf{\epsilon}_\theta$ is based on the public CSDI [1] code, and most of the architecture configurations follow CSDI such as variance schedules $\beta_t$ and $\alpha_t$ of diffusion.
> > > > For detailed values of the critical hyper-parameters, please see the table below.
> > > > Illustrations of model architectures are presented in Figure 3 in our main paper and Figures 9 and 10 in Appendix H.
> > > >
> > > > Here, we summarize the key components of the model architectures.
> > > > Given a time series $\mathbf{x}\_t$, our model will first slices $\mathbf{x}\_t$ into sequences with $R$ different resolutions (with different patch lengths) $\\{\mathbf{P}^r\\}\_{r=0}^{R-1}$.
> > > > Each sequence $\mathbf{P}^r$ will be fed into the processing module to predict the noise of $\mathbf{P}^r$.
> > > > We implement two versions of processing modules TEdit-CSDI and TEdit-TW based on CSDI [1] and Time Weaver [4], respectively.
> > > > Detailed figures, dimension sizes, and operations are presented in Figures 9 and 10 in Appendix H.
> > > > Our modified parts are highlighted in red in Figures 9 and 10.
> > > > Briefly speaking, CSDI does not take attributes as inputs and TEdit-CSDI incorporates attributes as prompt embeddings used in [5].
> > > > For Time Weaver, it is also based on CSDI and uses another strategy to incorporate attribute information.
> > > > We only make a slight modification by replacing the self-attention for processing continuous attributes with MLPs, since we do not consider continuous attributes in the current version.
> > > >
> > > >
> > > > ## 3. Hyper-parameters
> > > >
> > > > There are two types of hyper-parameters: model parameters and training parameters, which are listed below.
> > > > Many model parameters follow the settings of CSDI [1].
> > > > In the revised version, we will add more details.
> > > >
> > > > - **Table 1: model parameters**
> > > >
> > > > |Hyper-parameters|Descriptions|Values|
> > > > |:------:|:------:|:------:|
> > > > |$d$|Model dimension|128|
> > > > |$N$|The number of layers of the noise estimator|3|
> > > > |$R$|The number of resolution|2,3,4|
> > > > |$L_P$|Patch length|2,3|
> > > > |$T$|Diffusion steps|50|
> > > > |$S$|The start value for variance schedule|0.0001|
> > > > |$E$|The end value for variance schedule|0.5|
> > > > |$\beta_t$|Variance schedule for diffusion|$[\frac{S-E}{T-1}+S*(t-1)]^2$|
> > > >
> > > > - **Table 2: training parameters**
> > > >
> > > > |Hyper-parameters|Descriptions|Values|
> > > > |:------:|:------:|:------:|
> > > > |${E}_{pt}$|The epoch number of pretraining|300~1000|
> > > > |${lr}_{pt}$|The learning rate of pretraining|1e-3|
> > > > |${E}_{ft}$|The epoch number of finetuning|50|
> > > > |${lr}_{ft}$|The learning rate of finetuning|1e-5~1e-7|
> > > > |$\psi$|Bootstrap ratio|0.0~1.0|
> > > > |$\omega$|Bootstrap weight|0.0~1.0|
> > > >
> > > > [1] Tashiro et al. CSDI: Conditional score-based diffusion models for probabilistic time series imputation. NeurIPS'2021
> > > >
> > > > [2] Radford et al. Learning transferable visual models from natural language supervision. ICML'2021.
> > > >
> > > > [3] Nie et al. A Time Series is Worth 64 Words: Long-term Forecasting with Transformers. ICLR'2023
> > > >
> > > > [4] Narasimhan et al. Time weaver: A conditional time series generation model. ICML'2024
> > > >
> > > > [5] Jia et al. Visual prompt tuning. ECCV'2022

---

### Official Review · Reviewer_tfFr · 2024-07-12

**Soundness:** 4
**Presentation:** 3
**Contribution:** 2
**Rating:** 6
**Confidence:** 4

**Summary:**

This work proposes a new task as time series editing (TSE), that edits existing time series samples based on given sets of attributes. The goal is to perform edits that are only relevant to the changed attributes while preserving other existing attributes. The authors propose a multi-resolution modeling and generation model, and they perform bootstrap tuning on self-generated instances to perform the experiments. The authors show the proposed method can be used for modifying specified time series on synthetic dataset and real-world datasets.

**Strengths:**

- The idea is interesting. The TS editing could be an important task for time series.
- The proposed framework makes sense. The Multi-Resolution backbone clearly addresses issues of existing backbones; and the bootstrapped training method seems legit given the challenge of existing data/information.
- Experiments on synthetic datasets showed clear advantage of the proposed method.

**Weaknesses:**

While I personally like the idea a lot, there are a few non-neglectable issues (rooms for improvement):

1. The work could use better applications.
There are a couple of tasks/applications that could clearly benefit from TS editing. For example, see the works below:

[1] Cheng, J. Y., Goh, H., Dogrusoz, K., Tuzel, O., & Azemi, E. (2020). Subject-aware contrastive learning for biosignals. arXiv preprint arXiv:2007.04871.

[2] Liu, R., Azabou, M., Dabagia, M., Lin, C. H., Gheshlaghi Azar, M., Hengen, K., ... & Dyer, E. (2021). Drop, swap, and generate: A self-supervised approach for generating neural activity. Advances in neural information processing systems, 34, 10587-10599.

[3] Yi, K., Wang, Y., Ren, K., & Li, D. (2024). Learning topology-agnostic eeg representations with geometry-aware modeling. Advances in Neural Information Processing Systems, 36.

The above works include datasets with very specific attributes, e.g. subject information; task information; geo/channel information. If the proposed method could be applied to the above tasks, and demonstrate robustness, the paper could be much stronger.

2. Experimental results are overall not convincing enough. The datasets are too small, and there is no enough information about the synthetic dataset. There are not a lot of baselines.

3. Some parts of the method section can use better presentation:
- Section 3.1 How does the Time Series Editing task differ from the normal editing task in vision? Why do the authors want to define a new definition for it?
- Section 3.2 The authors should argue why is diffusion necessary for performing the task. What advantages would it bring?
- Section 3.4 The authors should compare the proposed method with traditional SSL methods like BYOL & Simclr and discuss about their differences.

**Questions:**

NA

**Limitations:**

The authors discussed limitations.

---

> ### Author Rebuttal · Authors · 2024-08-07
>
> Thanks for your appreciation and comments!
>
> >**W1. Suggested new datasets.**
>
> Thanks for your suggestion. The datasets you mentioned are indeed related and useful, revealing a wider application space for our work.
>
> We have looked into these datasets, many of which are EEG time series with various attributes, such as subject and task information. For example, the SEED dataset used in [1] contains attributes such as EEG channels and subjects, which is similar to the Motor dataset (Sec. 4.1 and Appendix E) used in our paper. Potential applications of these datasets include completing time series of missing channels by editing time series of other channels. Recall that, time series editing is a relatively new task, and there is still much to explore.  We plan to further investigate the evaluation benchmark of Time Series Editing (TSE) task and will use a broader range of datasets in future research.
>
> >**W2. Dataset size, synthetic data details and baselines.**
>
> Thanks for your comments.
>
> 1. For the dataset size, we believe the size should not be considered "too small" since the datasets we used and many datasets used in recent studies, e.g., time series generation [2] and LLM based time series model [3], have approximately the same scale. In total, our Synthetic-1/Synthetic-2/Motor/Air have 10,660/13,680/12,800/6,850 samples. Many datasets used in [2] and [3] range between 10,000~30,000. Collecting larger datasets will be an important future work for our paper and the proposed TSE task, and we will keep gathering more datasets with larger scales.
>
> 2. For the synthetic dataset, due to space limitation in the main paper, its details are presented in Appendix E.1.
> Briefly speaking, the synthetic samples we constructed consist of four parts: *trend*, *season*, *noise* and *bias*.
> - The *trend* has two attributes: trend types (linear, quadratic, exponential, logistic) and trend directions (up, down).
> - The attribute of *season* is the number of cycles, with possible values of 0, 1, 2, 4.
> - The *noise* is a combination of Gaussian noise and high-frequency noise.
> - The *bias* of each sample is randomly sampled from a uniform distribution.
>
> 3. Regarding the baselines, we are exploring a brand new field, and to the best of our knowledge, this is the first work focusing on developing methods and evaluation benchmarks for TSE. There are only few related works, and adapting the models from other tasks to this one involves significant efforts. The baselines we selected are Time Weaver [2], the latest work of conditional time series generation, and CSDI [4], a classic time series imputation framework.
> Please note that the latest work [2] only compared with 2 traditional GAN-based methods due to the lack of proper baselines in conditional time series generation. We believe [2][4] effectively reflect the current state-of-the-art in related fields.
> However, we will consider adding more comparative analyses in the future. Thank you for the suggestion.
>
> > **W3. Presentation suggestions.**
>
> Thank you for your suggestions and we will update the revised version accordingly for better readability.
>
> 1. About time series editing (TSE) and image editing.
>
> 1.1. Comparison of TSE and image editing.
>
> Broadly speaking, TSE shares the same target with image editing: edit the given data according to the specified conditions.
> However, time series data is quite different from image data, and there are unique challenges for TSE.
> - Time series has attributes such as *recoding location of city*, which are more difficult for model to interpret than the attributes such as *color* in image.
> - Time series data has its own unique properties, e.g., variant seasonality and low signal-to-noise ratio.
> - Moreover, the data of the time series corresponding to some specific conditions might be rare, making generation and editing quite challenging.
>
> Note that, we propose to use multi-resolution modeling and generation with bootstrapping mechanism trying to resolve the problem of aforementioned problems.
>
> 1.2 Definition presented in Section 3.1
>
> Though sharing similar definition of image editing task, in Section 3.1, we formally formulate TSE task mainly because editing is a relatively new concept for the time series research community. A formal problem formulation could help readers to easily understand the task and follow the paper.
>
> 2. The necessity of diffusion.
>
> TSE is a kind of generation task, which requires powerful generative models. There are two considerations of using diffusion models. (1) Diffusion is the latest generation technique and some recent works [2] have demonstrated that diffusion models can significantly outperform traditional methods like Generative Adversarial Network (GAN). (2) To keep fair comparison to the latest conditional time series generation work [2], we leverage the backbone model and base generation algorithm as the same as the baseline works [2][4].
>
> 3. BYOL and SimCLR
>
> The work BYOL you mentioned bootstraps the output of a neural network to serve as targets for a learning network to approximate. It only bootstraps on the learning targets and it is different from our method that utilizes the learned generative model to generate the samples of both input and targets for bootstrapping the learning of the target generative model. SimCLR adopts different data augmentation to build the target representation for contrastive learning, which does not leverage bootstrapping.
> We will add more discussions to other bootstrapping works to enhance the presentation of our work.
>
> [1] Yi et al. Learning topology-agnostic eeg representations with geometry-aware modeling. NeurIPS'2023
>
> [2] Narasimhan et al. Time weaver: A conditional time series generation model. ICML'2024
>
> [2] Jin et al, Time-LLM: Time Series Forecasting by Reprogramming Large Language Models. ICLR'2024
>
> [4] Tashiro et al. CSDI: Conditional score-based diffusion models for probabilistic time series imputation. NeurIPS'2021

---

> > ### Author Response · Authors · 2024-08-13
> > **Look forward to hearing from you**
> >
> > Dear Reviewer tfFr,
> >
> > Thanks for your valuable comments and feedback!
> > We hope our response could help address your concerns.
> > Today is the last day of discussion. We look forward to hearing from you and hope to discuss any concerns or suggestions to further improve our work.
> >
> > Thank you!
> >
> > Best,
> >
> > Authors of the 10940 submission

---

> ### Comment · Reviewer_tfFr · 2024-08-13
> **Thanks**
>
> Overall I like the work, especially the possibilities it brings to the field. I'd suggest the authors include more discussions during revision, especially regarding the applications that I suggested to encourage future follow-ups. The rebuttals also provided some other evidence for me to further advocate for the work and thus I will raise my score.

---

> > ### Author Response · Authors · 2024-08-13
> >
> > Thank you for your feedback!
> >
> > We will revise our paper according to your suggestions.

---

### Author Rebuttal · Authors · 2024-08-07

# General Response

We thank all the reviewers' insightful and valuable feedback!

We are encouraged by the positive comments from the reviewers such as:
- The proposed Time Series Editing (TSE) is an interesting and important task with a wide range of applications (Reviewer tfFr, ZaGr, EFAk and FqWw).
- The proposed framework TEdit, including bootstrapping and multi-resolution mechanism, is technically sound (Reviewer tfFr, ZaGr, EFAk, FqWw).
- Experimental results demonstrate the advantages of the proposed TEdit method as a new baseline (Reviewer tfFr, ZaGr, EFAk).
- Paper is clearly written and well-structured (Reviewer ZaGr).

Here, we also try to address some common concerns raised by the reviewers.

**Q1. Averaged results of different modifying attributes.**

As raised by reviewers ZaGr, EFAk, FqWw, the average value of modifying different attributes should be used as a more robust evaluation metric.
***We have provided the corresponding results in Tables 1 and 2, and also put the detailed experimental results of editing different attributes in the below attached PDF file in this General Response.*** Due to the time limit of rebuttal, we compared the performance of Time Weaver and our method on Synthetic dataset and Air real-world dataset. We will conduct full experiments in the revised version of our paper.

|Method|MSE$\downarrow$|MAE$\downarrow$|TAP_trend_type$\uparrow$|TAP_trend_direction$\uparrow$|TAP_season_cycle$\uparrow$|RaTS_edited$\uparrow$|RaTS_abs_preserved$\downarrow$|
|:--------:|:--------:|:--------:|:--------:|:---------:|:--------:|:---------:|:--------:|
|Time Weaver|0.1502|0.2953|0.6441|0.9856|0.5760|0.8537|0.1206|
|TEdit-TW (ours)|**0.1184**|**0.2641**|**0.7125**|**0.9870**|**0.7122**|**0.9599**|**0.1069**|
#### Table 1. Average performance of editing different attributes on Synthetic dataset.

|Method|TAP_city$\uparrow$|TAP_season$\uparrow$|RaTS_edited$\uparrow$|RaTS_preserved$\downarrow$|
|:--------:|:--------:|:--------:|:--------:|:--------:|
|Time Weaver|0.6386|0.2955|0.8799|0.2017|
|TEdit-TW (ours)|**0.7530**|**0.3303**|**0.9970**|**0.1753**|
#### Table 2. Average performance of editing different attributes on Air dataset.

From the above tables of average performance, our proposed method (TEdit) outperformed the baseline in all aspects on Synthetic dataset and real-world Air dataset.

However, averaging the performance on editing different attributes of time series neglects important insights on time series editing task.
From the detailed experiment results (as listed in the PDF file below), we found that the difficulty of editing different attributes is significantly different, and the average result may discard the difference. Specifically, we conclude the observations of the detailed performance on editing different attributes of time series as below.
* On both Synthetic dataset and real-world Air dataset, our method performs better than the baseline in almost all the settings.
* On Synthetic dataset, we notice that editing the attribute "trend type" is the easiest (better performance) while editing "season_cycle" is the most difficult.
* The difficulty of modifying multiple attributes is higher than that of modifying an individual attribute.
* On Air dataset, our method far exceeds the performance of the baseline in terms of the metrics of editing attributes.

We will add these discussions in the revised paper accordingly.


**Q2. About the baselines and related works.**

Some reviewers have recommended some other related works for discussion, including wider applications on real-world datasets (Reviewer tfFr) and more discussion on differentiation to the other works (Reviewer EFAk). These suggestions indeed help improve our work.

We briefly explain the baseline selection of our paper. As is been recognized by all the reviewers, the proposed task of time series editing is novel, while there are very few related works on this new direction. Our work compares two baselines, namely Time Weaver [1] which focuses on conditional generation of time series, and CSDI [2] which focuses on time series imputation tasks. Note that, these works are either the most cutting-edge research [1], or the classic yet pioneering work [2] in their respective fields. Both are representative in the latest research frontier in time series generation.

The works mentioned by Reviewer tfFr cover more datasets in addition to that in our paper. These new datasets extends the application of our work, and they also share similar processing pipeline to that used in our paper. We plan to leverage those new datasets to further enhance the benchmark of the proposed time series editing task, as detailed in the response of W1 of Reviewer tfFr.

Reviewer EFAk has provided some related works about time series generation and the model architecture. We have detailed the discussion of differentiation to these works in the response to W1 and W3 of Reviewer EFAk. We will add the discussion of these works in our revised paper.

[1] Narasimhan, Sai Shankar, et al. "Time weaver: A conditional time series generation model." ICML'2024.

[2] Tashiro, Yusuke, et al. "Csdi: Conditional score-based diffusion models for probabilistic time series imputation." NeurIPS'2021.

---

### Decision · Program_Chairs · 2024-09-25

**Decision:**

Accept (poster)

**Comment:**

This paper presents a novel approach to time series editing (TSE), a critical advancement in time series data synthesis. Unlike traditional methods that generate time series from scratch, which may lead to unexpected outcomes or trivial attribute prioritization, this work focuses on editing existing time series data to match specified attributes while preserving others. The authors propose a multi-resolution modeling framework, combined with bootstrap tuning on self-generated instances, to achieve this. The experiments demonstrate that this method is effective on both synthetic and real-world datasets, showcasing its potential to directly modify attributes while maintaining data integrity.

The significance of this work lies in its practical applicability and the innovation it brings to time series analysis. By enabling controlled edits on time series data, the proposed method addresses key challenges such as data sparsity and bias in time series generation. This paper not only offers a robust solution for generating high-quality time series data but also lays the groundwork for future research in domains where precise control over data attributes is necessary. The combination of a strong methodological foundation, comprehensive experimentation, and practical relevance makes this paper a valuable contribution to the field.